# Mission-Critical Services in 4G/5G and Beyond: Standardization, Key Challenges, and Future Perspectives

**DOI:** 10.3390/s25165156

**Published:** 2025-08-19

**Authors:** Florin Rastoceanu, Constantin Grozea, Mihai Enache, Raluca Nelega, Gergo Kovacs, Emanuel Puschita

**Affiliations:** 1Military Equipment and Technologies Research Agency, 16 Aeroportului Street, 077025 Clinceni, Romania; cgrozea@acttm.ro (C.G.); menache@acttm.ro (M.E.); 2National Institute for Research and Development of Isotopic and Molecular Technologies, 67-103 Donat Street, 400293 Cluj-Napoca, Romania; raluca.nelega@itim-cj.ro (R.N.); gergo.kovacs@itim-cj.ro (G.K.); 3Communications Department, Technical University of Cluj-Napoca, 26-28 George Barițiu Street, 400027 Cluj-Napoca, Romania

**Keywords:** mission-critical services (MCX), LTE/4G/5G networks, LTE, 3GPP standardization, Quality of Service (QoS), Public Protection and Disaster Relief (PPDR), Ultra-Reliable Low-Latency Communications (URLLC), Proximity Services (ProSe)

## Abstract

Mission-critical services (MCX) comprise a standardized suite of capabilities including Mission-Critical Push-to-Talk (MCPTT), MCVideo, and MCData, designed to meet stringent requirements for availability, reliability, latency, security, and Quality of Service (QoS). These services are essential for public safety, emergency response, and other critical infrastructure domains, where communication performance directly affects operational effectiveness. Integration into 4G and 5G mobile networks, supported by targeted standardization efforts, has extended broadband capabilities to mission-critical environments. 5G networks provide the technical foundations for MCX through ultra-low latency (below 1 ms), high availability (99.999%), broadband throughput over 100 Mbps per user, deterministic QoS via network slicing, massive device connectivity (over one million devices per square kilometer), and seamless Non-Terrestrial Network (NTN) integration. Technical enablers such as Proximity Services (ProSe), network slicing, and Ultra-Reliable Low-Latency Communications (URLLC) are fundamental to delivering these capabilities. This paper reviews MCX architectures, service frameworks, and protocols, relating MCPTT, MCData, and MCVideo to the key performance requirements defined in ITU-T M.2377-2. It also examines the frozen features of 3GPP Release 19, including enhancements to MC services, NTN integration, Reduced Capability device support, sub-meter positioning, extended network slicing for Public Protection and Disaster Relief (PPDR), and strengthened security mechanisms. Finally, the study addresses challenges such as standard maturity, interoperability, and deterministic QoS, identifying research priorities toward 6G readiness. By consolidating advances from standards bodies, research initiatives, and deployments, this work serves as a technical reference for scalable, secure, and standards-compliant MCX solutions in current and future networks.

## 1. Introduction

Mission-critical services (MCX), as defined by 3GPP, represent a set of standardized communication functionalities—namely Mission-Critical Push-to-Talk (MCPTT), MCVideo, and MCData—designed to meet the stringent operational demands of public safety, emergency response, and other critical infrastructure domains. These services impose strict requirements on latency, availability, reliability, and security, exceeding those of conventional broadband services.

Over the last decade, mobile networks have evolved significantly, particularly with the advent of LTE and 5G, to support these critical requirements. In parallel, standardization bodies such as 3GPP, ETSI, and ITU have defined architectural frameworks and service specifications that enable MCX capabilities in IP-based mobile environments. Technologies like Proximity Services (ProSe), network slicing, and Ultra-Reliable Low-Latency Communications (URLLC) are among the key enablers introduced in 3GPP releases to support mission-critical use cases.

While 5G technology is still maturing and undergoing active development [1], early efforts to define the next generation—6G—have already begun. According to the ITU-R IMT 2030 vision for the 6G era [2], key advancements such as ultra-fast data transmission speeds, extreme latency reduction, and ubiquitous integrated sensing are expected to redefine how MCX services evolve.

Organizations such as 3GPP (through Release 19) and academic research initiatives [3,4] are already working on incorporating MCX into the future 6G service ecosystem [5]. Key enabling technologies include Non-Terrestrial Networks (NTNs) [6,7], expected to provide extended coverage in remote or disaster-affected areas, and artificial intelligence (AI) [8,9], which will support predictive analytics, network automation, and resilient decision-making for MCX.

Based on these developments, this paper sets out two main objectives: (1) to assess the current state of standardization, service features, and enablers supporting MCX in LTE/5G systems; and (2) to explore future research directions, including emerging technologies and their potential role in advancing MCX in the context of 6G.

While various studies and reports address individual components or service types within MCX, there is a limited number of consolidated works that offer a comprehensive and up-to-date overview of standardization efforts, enabling technologies, and the evolving research landscape. This paper aims to fill this gap by providing a structured and critical review of MCX support in 4G/5G networks, with a forward-looking perspective toward 6G integration.

The research approach was based on a structured literature review, conducted in three stages: first, identifying peer-reviewed works from databases such as IEEE Xplore and ScienceDirect (2020–2025) using MCX-related keywords; second, analyzing official standards and industrial sources including 3GPP, ITU-R, 5G-ACIA, and TCCA; and third, organizing the results thematically by standardization, architecture, enabling technologies, and vertical use cases.

The main contributions of this paper are as follows: (1) an in-depth synthesis of 3GPP standardization progress relevant to MCX services across LTE and 5G; (2) a technical examination of core architectural components and communication enablers such as ProSe, URLLC, and network slicing; (3) identification and analysis of persistent challenges in areas such as QoS, interoperability, and security; and (4) a perspective on ongoing research trends and future directions, particularly regarding MCX evolution in the context of 6G.

The remainder of the paper is organized as follows: Section 2 introduces the MCX standardization and current developments, Section 3 discusses current research trends and open challenges; and Section 4 concludes with future perspectives toward 6G.

## 2. MCX Standardization and Current Developments

### 2.1. MCX Standardization Efforts

Several concurrent standardization efforts are ongoing in the MCX field [10]. The most important work is being carried out by the 3GPP project, which included MCX in its standardization process from 2014. Additionally, ETSI has a dedicated workstream for this purpose. In parallel, the O-RAN Alliance is responsible for defining OpenRAN specifications, which are subsequently adopted by ETSI. Furthermore, the Global Certification Forum (GCF) collaborates with the Terrestrial Trunked Radio (TETRA) and TCCA to offer certification for broadband mission-critical services.

#### 2.1.1. 3GPP Standardization

3GPP is actively involved across all stages of the innovation cycle, including concept development, standardization, testing, and commercialization of mobile telecommunications systems. Its essential role is to specify and maintain complete system descriptions. This approach to standardization enables various implementations of mobile telecommunications systems. As a result, it supports interoperable, multi-vendor environments that promote economies of scale in mass production. At the same time, it allows for other innovations in the field to develop freely [11]. Figure 1 presents the role of 3GPP in the standardization process.

Participation in 3GPP is made possible by companies and organizations becoming members of one of the 3GPP Organizational Partners, the seven Standards Developing Organizations (SDOs)—from Europe, China, Japan, India, Korea and the United States. Specific contributions, in the form of market requirements, may also come into the project through any of the more than twenty Market Representation Partners (MRPs) in 3GPP. These organizations have all signed up to the 3GPP Project scope and objectives. There is also a lot of external cooperation with other standards bodies and a broad variety of other groups (companies, government departments, educational establishments, etc.), by way of formal liaisons [11].

SA6—Application Enablement and Critical Communication work group has the lead for mission-critical applications in the 3GPP standardization process. Its work started in 2014 with version 13 and since then it has been involved in MCX standardization efforts. The evolution of MCX standardization is presented in Table 1.

Significant work related to mission-critical services was made by other groups. Figure 2 illustrates the roles of various 3GPP working groups, including SA6, SA4, SA3, SA1 and CT1, in defining MCX service specifications.

#### 2.1.2. ETSI Standards

ETSI is a non-profit organization with a major role in European Information and Communication Technology (ICT) standardization processes. Two of the committees have, as their main objective, the standardization of the field of MCX. The first one is the TETRA and Critical Communications Evolution (TCCE) committee, and its main objective is to maintain and develop new TETRA standards for mission-critical communications. The standardization efforts include security measures through encryption updates and integration with modern technologies, such as mission-critical broadband systems, to support mission-critical operational requirements.

The second committee is TC Emergency Communications committee that focuses on developing Next Generation 112 (NG112) standards to modernize emergency services communication. Its standards enable citizens to contact emergency services using everyday communication technologies, including multimedia capabilities. Key deliverables include the EMTEL TS 103 479 [12], which defines an IP-based architecture with core elements and technical interfaces, and the ETSI TS 103 479 [13], which addresses the accessibility and interoperability of emergency communications.

#### 2.1.3. O-Ran Alliance

The O-RAN Alliance, founded in 2018 by major telecom operators, has evolved into a global community focused on reshaping the Radio Access Network (RAN) industry. Its mission is to develop open, intelligent, fully interoperable virtualized mobile networks. By creating specifications that enable a competitive and innovative RAN supplier ecosystem, the alliance improves user experience and accelerates innovation.

The alliance’s efforts regarding open and interoperable networks facilitate the implementation of MCX services. This support allows for the development of more flexible and efficient network architectures, which are capable of meeting the strict requirements of critical communications.

#### 2.1.4. Global Certification Forum (GCF)

The Global Certification Forum and the TETRA and Critical Communications Association (TCCA) have jointly developed a certification program for MCX, focusing on 3GPP-based services, frequency bands, and functionalities prioritized by the mission-critical industry. In June 2024, the GCF began a certification program in collaboration with TCCA for Broadband Mission Critical Services [14]. 3GPP standards regarding MCX are included in its certification program. MCPTT from Release 14 was first included, continuing with other MCX services, such as MCData and MCVideo, as well as newer 3GPP releases. The certification plan is to be extended to MCX servers, covering both server-to-client and server-to-server interfaces.

#### 2.1.5. BroadEU.Net—A European Project

BroadEU.Net is a partnership involving 16 EU and Schengen member states. Its main goal is to organize the EU Critical Communication System (EUCCS) by 2030. This project’s main objective is to standardize and integrate MCX across Europe, continuing the work conducted in the BroadWay project. Technical and operational aspects were taken into account in this process. These have included tests performed jointly by nations deploying different types of MCXs and new operational procedures to improve the level of operability between European nations.

The standardization effort addresses the operational needs for server-to-server mission-critical communications by taking into account the technical challenges encountered in the different testing phases. The governance, policy, and legislative issues are driven by the European Commission’s Mission Critical Expert Group (MCCG). BroadEU.Net aims to develop a unified critical communication system for emergency responders across Europe. To achieve this, it involves the entire ecosystem, including member states, industry, and research communities. The system will be based on 3GPP standards and specifically tailored to meet operational needs [10].

### 2.2. Mission-Critical Services Developments

The standardization effort for MCX made by 3GPP has been essential in the transition of critical communications from traditional networks (TETRA, P25) to Long-Term Evolution (LTE) and 5G networks. The evolution of technologies used during the evolution of MCX standardization is presented in Table 2.

#### 2.2.1. MCPTT

Mission-Critical Push-To-Talk (MCPTT) is a standardized service by 3GPP that enables reliable, instant, and prioritized communication for public safety organizations over LTE and 5G networks [51,52]. MCPTT is designed to enable both one-to-one and one-to-many communications. It aims to replicate the capabilities of traditional land-mobile radio (LMR) systems, while taking advantage of the wide coverage provided by IP-based networks. MCPTT forms a critical component of the broader IP-based PTT ecosystem. PTT involves an application installed on end-user mobile devices. This application communicates over cellular data networks, typically using VoIP over LTE or 5G, and relies on IMS-based infrastructure. Audio, data, telemetry, and location services are transmitted through the Radio Access Network (RAN) to a central PTT server. This server can be hosted by a cellular provider or operated in the cloud by application service providers. Functionally, PTT extends the typical capabilities of smartphones. For mission-critical applications, the core system is often isolated from the carrier’s mainstream networks and connected via virtual private networks (VPN) [53].

The introduction of the basic requirements for MCPTT was adopted in Rel-12 (2015). The evolution of MCPTT service characteristics continued with the development of comprehensive specifications. These specifications were adopted in Release 13 (2016) and were designed to address the specific and complex requirements of emergency and public safety services. A number of improvements to MCPTT were adopted in Rel-14 (2017), Rel-15 (2018), Rel-16 (2020), Rel-17 (2022) and work was conducted to integrate MCPTT with other critical services, such as MCData and MCVideo, to create an integrated communications ecosystem. The full integration of MCPTT into MCX services, in a communications ecosystem with MCData and MCVideo, was adopted in Rel-18 (2024) and will be enhanced in Rel-19 (in progress) (see also Table 1).

In research papers, MCPTT services are examined from various perspectives. These include device-to-device communication, signal-to-noise ratio (SNR) analysis, prediction and evaluation of key performance indicators (KPIs), mission-critical Internet of Things (IoT), transmission protocols, broadcast transmission modes, and applications in public safety and emergency scenarios. In [54], an efficient MCPTT supported concurrent conversations serialization of voice streams at the receiver. A device-to-device communication was proposed [55]. The study [56] analyzed the role of wearable devices with cellular support in Internet of Life Saving Things use cases and conducted a performance evaluation in MCPTT applications using smartwatches with LTE Cat-M2 support. An analysis of the performance of four scenarios by simulation and SNR evaluation using Rayleigh fading incorporating path loss model is realized in [57]. The study [58] proposed a heterogeneous network handover algorithm for hybrid networking mode to be used in a MCPTT system. A comparative study for LMR and LTE networks highlighting the trade-offs in the context of MCPTT used by emergency first responders is performed in [59]. Evaluation of 5G-enabled MCPTT services used in a city’s smart infrastructure is realized in [60]. The findings of the study show that 5G outperforms LMR and 4G in terms of decreased blocking likelihood and shorter waiting times. In [61], a solution is described that extends the ns-3 NR V2X simulator capabilities for MCPTT use to cater to New Radio ProSe public safety applications.

#### 2.2.2. MCData

MCData is a standardized service by 3GPP that allows users to send and receive data in real time, including large files, images, videos, or other types of data needed in critical scenarios networks [62,63]. MCData transfers occur over LTE and 5G networks, leveraging their enhanced bandwidth and reliability. MCData enables users to utilize data communication functionalities, such as messaging, file distribution, and data streaming. These features facilitate quick decision-making by providing decision-makers with real-time access to information in the form of data. As a critical communications service for public safety and emergency response agencies, MCData guarantees fast and reliable data transfers, even under congested network conditions. For example, in the event of accidents or emergencies, agencies can quickly transmit field images or other essential data to coordinate intervention actions. MCData is interoperable with other critical communications services, including MCPTT and MCVideo. This is made through the MCX plugtests and was tested by ETSI. This enables efficient integration of voice, video, and data communications, creating a unified ecosystem for emergency response organizations. Within MCData, users can form communication groups to send data to all members or to selected individuals. This is important in emergency situations, when response agencies need to quickly share crucial information in a coordinated manner. MCData is also important for integration with emergency resource management systems and relevant data. An important aspect of MCData is the security of the transmitted information, given the critical nature of the data. All communications are encrypted and protected against unauthorized access.

The evolution of the MCData service features took place starting with the development of full specifications for MCData. This was adopted in Rel-14 (2017). A number of improvements to MCData were adopted in Rel-15 (2018), Rel-16 (2020), Rel-17 (2022) and work was conducted to integrate MCData with other critical services such as MCPTT and MCVideo to create an integrated communications ecosystem. The full integration of MCData into MCX services, in a communications ecosystem with MCPTT and MCVideo, was adopted in Rel-18 (2024) and will be enhanced in Rel-19 (in progress) (see also Table 1).

In research papers, MCData services are addressed in terms of Ultra-Reliable and Low-Latency Communication (URLLC), KPI prediction, and data transmission protocol. The study [64] proposed solutions for secure URLLC in mission-critical Internet of Things. In [65], a digital twin to predict KPIs such as data rate and latency for mission-critical applications was proposed and tested on real a case study and in [66], several multi-path-based solutions for IoT data transmissions protocols were proposed.

#### 2.2.3. MCVideo

MCVideo is a 3GPP standardized service for facilitating real-time video transmissions between public safety agencies and emergency response teams networks [67,68]. MCVideo allows users in the field of public safety to use video communication capabilities through group and private video calls. Critical situation management at the decision-maker level is improved by elevating communication priorities. This is conducted by inserting flags for urgent video communications, enabling live video streaming, remotely sharing videos on compatible devices, and transmitting video excerpts as needed. In a critical situation, visual information is an essential service for critical communications, which can be as important as voice communication. MCVideo provides a reliable and fast framework for exchanging video images, supporting the efficient coordination of intervention actions. MCVideo allows public safety agencies and response teams to transmit and receive high-quality video streams in real time. This is crucial in emergency scenarios, where visual assessment of a situation can make a difference in response planning. MCVideo not only allows one-way transmissions (for example, sending a video stream to the coordination center), but also two-way video, which means that agents in the field can receive visual guidance and communicate live via video with remote teams. MCVideo uses LTE and 5G networks to ensure quality video streams, even in limited bandwidth conditions. Combining this service with 4G or 5G technology allows for the transmission of high-resolution videos, essential for public safety and rapid response missions. In emergency scenarios, MCVideo benefits from network prioritization, so that emergency video streams are handled with priority over less important data traffic. This ensures that response agencies can access essential video streams without delay, even under network congestion. MCVideo is fully interoperable with other mission-critical services, such as MCPTT and MCData. This interoperability allows response teams to communicate with voice, exchange data, and transmit videos in an integrated and efficient manner. Given the nature of the information transmitted in critical situations, MCVideo includes advanced security measures, such as video stream encryption and user authentication, to protect the confidentiality and integrity of the transmitted data.

The development of full specifications for MCVideo was adopted in Rel-14 (2017). A number of enhancements to MCVideo were adopted in Rel-15 (2018) and work was underway to integrate MCVideo with other critical services such as MCPTT and MCData to create an integrated communications ecosystem. The full integration of MCVideo into MCX services, in a communications ecosystem with MCPTT and MCData, was adopted in Rel-18 (2024) and will be enhanced in Rel-19 (in progress) (see also Table 1).

In research papers, MCVideo is addressed in terms of fault management, video trunking communication and broadcast transmission. Paper [69] proposed a fault management window algorithm based on dynamic fault information feedback to reduce the large load on the MCVideo systems and paper [70] proposed dual-module communication and retransmission strategies to optimize reliability of MCVideo services implemented in Long-Term Evolution for railway systems. Study [71] compared, in terms of resources utilized, two broadcast transmission modes proposed in Multimedia Broadcast Multicast Service to be used in video streaming: Multicast Broadcast Single Frequency Network (MBSFN) and Single-Cell Point-to-Multipoint (SC-PTM). In [72], a multi-UAV-assisted public safety communication network for video transmissions from affected areas’ ground users in the case of emergency scenarios was proposed.

#### 2.2.4. Additional Services

In addition to the core critical communications services already mentioned, additional but essential services have been introduced to support critical communications, particularly for public safety and emergency response applications. These additional services (e.g., railways and MCIOPS) are designed to improve the ability of public safety agencies and response teams to collaborate and respond quickly and effectively in various critical scenarios (see also Table 1).

MC Interworking (migration and interconnection) was adopted in Rel-15 (2018) for migration to another MC system. It takes part in communications with users in another MC system (interconnection). Railways were introduced in Release 15 (2018) to enable the use of functional aliases and the multi-talker feature. This capability was further improved in Release 18 (2024), which added interconnection and migration aspects. The MBMS API for MCX was introduced in Release 16 (2020) to establish a reference model. This model leverages MBMS APIs to enable multicast mission-critical services. Furthermore, UE APIs are included to provide access to these MBMS mission-critical services specifically for mission-critical applications. MCIOPS was adopted in Rel-17 (2022) for the case of a backhaul failure or a nomadic EPS deployment; MC services shall be supported based on the availability of an Isolated E-UTRAN operation for Public Safety (IOPS) system. MC gateway UE was adopted in Rel-18 (2024) and enables MC service access for MC users using non-3GPP devices (which may or may not have the ability to host MC clients). MCAHGC was adopted in Rel-18 (2024). An ad-hoc group is set up spontaneously based on a list of users or some criteria (e.g., location) [73].

In [74], a study on location enhancements for mission-critical services (MC Location) was presented. Another study, focused on discreet listening and logging for mission-critical services (MC Logging), outlined solutions for discreet listening and recording in MC services [75]. The Mission Critical services over 5G System (MCOver5GS) explores how to use the 5G System (5GS) for MC services [76]. Additionally, the Gateway User Equipment (UE) function for Mission-Critical communications (MGWUE) study presented solutions to fulfill the requirements of a mission-critical EU Gateway function [77]. The introduction of the MC gateway UE in Release 18 (2024) facilitates MC service access for MC users using non-3GPP devices, even if these devices do not host MC clients [73]. Furthermore, Mission Critical Services over 5MBS (MCOver5MBS) addressed transport capacities in multicast and broadcast communications in networks for mission-critical services via 5GS [78]. Lastly, Mission Critical Services over 5GProSe (MCOver5GproSe) discussed similar transport capacities using 5G ProSe [79].

### 2.3. MCX Service Frameworks and Vertical Enablers

According to reference [80], one of the objectives of the SA6 workgroup is to develop application layer architecture specifications for 3GPP verticals. This includes critical communications applications, service frameworks, and vertical application enablers, as illustrated in Table 1.

According to reference [81], 5G service frameworks enhance and simplify the capabilities of the underlying 5G service system. They facilitate the development of third-party applications and enable a quicker incorporation and deployment of new vertical services over 3GPP networks. 5G Service Frameworks implemented in 3GPP and is presented below.

Common API Framework (CAPIF) is designed as a unified Application Programming Interface (API) framework to serve both network and application functionalities and aims to facilitate a simplified and harmonized approach to API development within 3GPP. This unified approach ensures that vertical applications (also called API invokers) have single-entry points to common API aspects (also called CAPIF APIs), such as onboarding, discovery, authentication, and authorization [80]. CAPIF is also designed to create a unified and harmonized platform for APIs that are not specified by 3GPP. This includes APIs defined by organizations such as the ETSI Industry Specification Group (ISG) for Multi-access Edge Computing (MEC), TM Forum (which has over 800 member companies), CAMARA (an open-source project within the Linux Foundation focused on defining, developing, and testing APIs), and others.

Service Enabler Architecture Layer (SEAL) specifies the application plane and signaling plane entities associated with application-enabling services (e.g., configuration management, location management, group management, network resource management, identity/key management) and can thus be reused in vertical applications [80]. SEAL services are supported for both user equipment (UE) on-network and off-network in a business-to-business (B2B) model.

Enabling Edge Application Enablement (EDGEAPP) offers an edge enablement layer and an application architecture designed to support the development and deployment of edge applications within the edge data network. This includes exposing Nordic APIs for edge applications, allowing integration with the 3GPP network, and enabling communication between application clients on user devices and edge application servers deployed in the edge data network. By providing these functionalities, EDGEAPP facilitates new capabilities such as improved service delivery, enhanced application discovery, and seamless service continuity [81]. Enabling EDGEAPP increases the performance goals of verticals, providing low latency and massive bandwidth.

SEAL data delivery enabler for vertical applications (SEALDD) provides the layer platform architecture, capabilities, and services that enable application efficiency in content/data storage and delivery for vertical applications (as part of SEAL services) [82]. Application Data Analytics Enablement Service (ADAES) presents the protocol aspects of the ADAE of SEAL services, which specifies the UE that supports the client functionality of the ADAE SEAL services and the network that supports the server functionality of the ADAE SEAL services [83]. Network Slice Capability Exposure for Application Layer Enablement (NSCALE) specifies the procedures and information flows necessary for Network Slice Capability Exposure for Application Layer Enablement on the basis of SEAL [84]. 5G-enabled fused location service capability exposure (5GFLS) presents the function/mechanism for improving 5G location services in the application-enabled layer. Application enablement aspects for subscriber-aware northbound API access (SNAAPP) provides the means of authorizing API invokers when access to protected resources is granted by the owners of these resources [85]. Guidelines for CAPIF Usage (CAPIF_EXT) presents guidance and options for using CAPIF and describes the implementation in CAPIF [86]. The study [87] on application layer support for AI/ML services (AIMLAPP), is a technical report that aims to identify the appropriate architecture, capabilities, and services at the application layer. Its purpose is to support and enable application-level AI and ML services. Study on enhanced application layer support for location services (eLSApp) proposes solutions to improve location determination issues [88]. The study on application enablement for the Localized Mobile Metaverse (Metaverse_App) examines the architecture needed for activating metaverse mobile services over 3GPP networks. It identifies key challenges and proposes solutions to support the activation and deployment of metaverse applications effectively [89]. Application enablement for XRM Services Phase 2 (XRM_Ph2_App) presents a study on key issues, solutions and conclusions for supporting advanced media services, such as High Data Rate Low-Latency (HDRLL) services and AR/VR/XR services [90].

While 5G service frameworks provide particularly useful horizontal capabilities, SA6 also introduced vertical application-enabling initiatives that address vertical-specific application developers. These initiatives are designed to leverage the capabilities offered by 5G service frameworks by layering on top with application enablement verticals and then employing their resulting services. In addition, 5G vertical application enablers may offer other customized features specific to but meeting the requirements of individual verticals [81].

In 3GPP are implemented enablers for the following vertical applications: Vehicle-to-Everything (V2X), Uncrewed Aerial System (UAS), MSGin5G Service, Personal IoT Networks, 5G satellite access, and Multimedia Telephony.

Several vertical enablers are treated in 3GPP [80]. V2XAPP provides application layer support for Vehicle-to-Everything (V2X) services. It enables the efficient use and deployment of V2X applications over 3GPP systems, including features such as platooning, advanced remote driving, and high-definition maps. Additionally, V2XAPP describes the wireless communication between a vehicle and any other entity that could impact or be impacted by the vehicle. UASAPP is an application layer support for UAS that specifies application layer capabilities towards UAS applications. FFAPP provides application layer support for the “factories of the future.” It addresses key issues and offers solutions related to Quality of Service (QoS) coordination, group communication, clock synchronization, and integration with existing operational technologies. 5GMARCH is an application architecture for MSGin5G Service that offers 5G messaging communication capabilities for massive IoT, including point-to-point, application-to-point, and group and broadcast messaging, across multiple UE types.

Many APIs have been designed, evolved, and matured over the last few 3GPP releases. 3GPP APIs provide multiple and varied capabilities. These capabilities provide services in a comprehensive, hierarchical manner and provide numerous open paths for enabling diverse sets of third-party application use cases enabled on mobile networks.

According to [91], 5G Messaging service (MSGin5G Service) represents an update to the mobile SMS service, with the aim of communicating and accessing information between users in a more convenient and efficient way. Personal IoT Networks (PINAPP) is designed as application support for Personal IoT Networks (PIN). It includes advanced features to improve the functionality, security, and interoperability of personal IoT devices [92]. 5G satellite access (5GSAT_Ph3_APP) allows the activation of applications for 5G services delivered via satellite access [93]. Multimedia Telephony (MMTel_App) is a globally recognized standard built on the IP Multimedia Subsystem (IMS). It is a service standardized by both 3GPP and ETSI/TISPAN, designed to support real-time communication capabilities such as voice, video telephony, and real-time chat [94].

### 2.4. Mission-Critical Verticals

The concept of verticals is employed in mission-critical services to develop tailored solutions that meet the specific requirements and challenges of various industries, like public safety and emergency services, transportation, power grids, and resource industries. Including verticals in the system design ensures interoperability and standardization across different domains. This approach also enhances the optimization of network essential performance characteristics, such as low latency, high reliability, and high bandwidth. It also facilitates the integration of emerging technologies such as 5G/6G, Non-Terrestrial Networks (NTN), Artificial Intelligence (AI), and Internet of Things (IoT). This integration helps align these technologies with the specific needs and requirements of various industries. Moreover, the vertical concept takes into account the specific aspects of each industry, while managing not to lose operability.

#### 2.4.1. Public Safety and Emergency Services

Mission-critical services are fundamental for sustaining effective and reliable responses in public safety and emergency operations. Initially supported by narrowband systems such as Project 25 (P25) in North America [95] and Terrestrial Trunked Radio (TETRA) in Europe [96]—offering high availability but limited bandwidth—these communications evolved with the 3rd Generation Partnership Project (3GPP) Release 13 [97] to Long-Term Evolution (LTE) and Fifth Generation (5G) broadband capabilities, integrating Mission-Critical Push-to-Talk (MCPTT), Mission-Critical Video (MCVideo), and Mission-Critical Data (MCData). This transition enables low-latency, high-throughput, and multimedia-rich communication, essential for Public Protection and Disaster Relief (PPDR) operations. The adoption of push-to-X technologies in 3GPP standards extends capabilities to include private and group video calls, multi-device and multi-line voice communications, text-based messaging, media sharing, and presence checking, enhancing situational awareness through technologies such as Augmented Reality (AR) and Virtual Reality (VR).

PPDR mission-critical networks must meet stringent requirements defined in the International Telecommunication Union Telecommunication Standardization Sector (ITU-T) M.2377-2 report [98], covering reliability, availability, coverage, capacity, interoperability, security, and operational efficiency. Key Performance Indicators (KPI) include 99.99% availability (uptime), call setup time under 500 milliseconds, N+1 redundancy for core network components, geographic coverage of 99% of the designated region, minimum cell throughput of 256 kbps per user during peak load, successful inter-agency communication rates above 99%, 100% encryption of voice and data, and battery life exceeding eight hours in active operational use.

These parameters are detailed in Table 3.

#### 2.4.2. Transportation—Future Railway Mobile Communication System (FRMCS)

The Future Railway Mobile Communication System is a next-generation communication framework designed to modernize railway operations by replacing the aging GSM-R system. Using the 3GPP standard as a solid base, FRMCS offers high-speed, low-latency, and reliable communication essential for mission-critical applications such as train control, signaling, and real-time monitoring [99]. FRMCS is designed to address the unique challenges of railway environments. These include ensuring reliable coverage in tunnels, remote areas, and at high speeds. It aims to provide a standardized, interoperable, and future-proof communication solution for the global railway industry. Future advancements like autonomous train operations, real-time diagnostics, and predictive analytics can be implemented using 5G technology. Additionally, 3GPP standards facilitate enhanced passenger services, such as high-speed onboard Wi-Fi, infotainment systems, and seamless connectivity for passengers during their journeys. The robust and standardized communication framework provided by 3GPP ensures that these services are reliable and interoperable across different railway networks.

The future of railway systems is heading towards increased networking, intelligence, and automation. The next-generation railway communication system aims to enable comprehensive perception, interconnection, and seamless information exchange among all users and infrastructure components. Intelligent applications like video-based track monitoring, ultra-reliable train control, as well as the massive access of users and the high number of sensors align with 5G’s key scenarios: Enhanced Mobile Broadband, Ultra-Reliable Low-Latency Communication, and Massive Machine-Type Communication. With advanced technologies, 5G-R is poised to meet demanding communication needs of modern railways. According to [100] there are six key point technologies that address the future of FRMCS. The new network architecture that uses network slicing enables flexible, customizable services and resource-sharing platforms. Massive MIMO (Multiple Input Multiple Output) reduces channel fading and handover issues. In this way, the system capacity and reliability are enhanced, particularly for high-speed trains. Millimeter-Wave technology leverages abundant bandwidth for Gbps-level data rates. IoT technology integrates massive low-power sensors, enabling comprehensive situation awareness. Ultra-reliable low-latency communication ensures sub-100 ms latency and 99.9999% reliability for critical operations, even at 500 km/h. Lastly, advanced video processing employs deep learning for enhanced railway monitoring, security, and passenger flow analysis.

The scientific activity on FRMC is quite intense in the period analyzed by this survey. The general aspects addressed are as follows: handover procedure, latency, reliability, identity privacy, user authentication, antennas design, interference management and adaptation to 5G systems. The study [101] investigated the effect of handover procedures on latency, reliability and mobility in railway communication systems. The papers [102,103] proposed solutions for microstrip MIMO antennas to be used in future railroad mobile communications. In [104], a new 3-D wide band geometry-based stochastic model for analyzing cooperative massive MIMO channels in high-speed rail communication systems was proposed and in [105], a method to provide identity privacy of the functional identities in railway communication systems was studied. The paper [106] introduced methods for network planning for future communications in long-distance rail systems. The solution ensures efficient long-distance railway communications by solving the problems of base station and data center placements. An overview of the developments and integration effort of 5G technology in Chinese railways, focusing on network architecture and service applications, was presented in [107]. The paper [108] proposed a linear-cell-based radio-over-fiber system for efficient communication in high-speed trains, Ref. [109] investigated interference management schemes in FRMC, using the dark-blind interference alignment principle and [110] presented initial implementations and field evaluations o FRMCS networks, focusing on MCX integration with 5G standalone network.

#### 2.4.3. Power Grids

A power grid is an interconnected network that generates, transmits, and distributes electricity from producers to consumers. It is structured in a clear hierarchy with electricity generation at the top and end users at the bottom. At the top of this hierarchy, the voltage is very high, typically ranging between 200 and 400 kV, and decreases as it moves down the network. Current energy systems rely on large energy producers such as nuclear and hydro power plants. They also include smaller-scale producers, like small wind farms, solar parks, and individual homeowners installing solar panels on their roofs. Today, the communication network in smart grid systems can be conceptualized as a hierarchical multi-layer architecture, categorized by data rate and coverage range. According to [111], it is structured on three levels. The first one is the local level and comprises Home Area Networks (HAN), Building Area Networks (BAN), and Industrial Area Networks (IAN). The second level, the intermediate level, comprises more complex networks such as Neighborhood Area Networks (NAN) or Field Area Networks (FAN). The top level, which provides the largest coverage, links the different NAN and FAN networks to create an interconnected smart grid communication infrastructure. This layered approach enables efficient data management and communication across different scales of the power distribution system. The future of smart grids must interconnect all the producers, consumers, and other actors in a single communication infrastructure. The rising popularity of renewable and distributed power generation is driving the need for power grids to evolve significantly. These advanced systems need to support a wide range of decentralized power sources. They must enable bi-directional energy flow, allowing prosumers to both buy and sell electricity. Additionally, the systems should adapt to the increased variability of renewable energy production and handle a higher frequency of network issues that can impact power quality. This transformation is essential to successfully integrate sustainable energy sources while maintaining grid stability and reliability in an ever more complex energy landscape [112].

The typical topology of a power grid today and tomorrow is presented in Figure 3.

According to [113], the future electric grid demands a substantial architectural transformation to meet emerging requirements. The next-generation grid communications architecture should embody several key attributes:Reliability and resilience by maintaining functionality during natural disasters, power outages, storms, network disruptions, and cyber-attacks.Durability and flexibility—the architecture’s lifespan should match or exceed that of indoor power grid equipment, without requiring major modifications. It must be adaptable to accommodate new devices, applications, and requirements without wholesale restructuring that would negate existing investments.Security and privacy—the system should comprehensively understand network characteristics, connected devices, and communication pathways to effectively monitor for anomalies indicative of malicious activity. This must be achieved while maintaining grid functionality and performance.Interoperability and standards—to minimize risk and avoid vendor lock-in, the architecture should utilize standards-based technologies. It should be easily upgradeable and support mix-and-match capabilities for best-of-breed equipment installation.Performance and scalability—the system must support applications with strict Quality of Service (QoS) requirements. The architecture must be capable of connecting a large number of endpoints and diverse networks. The process of transforming current electric power grids to smart grids makes use of emerging technologies.

Future research directions in this field were addressed in the following scientific papers. The main research directions were directed in the following areas of interest: 5G network slicing, network quality, resource allocation, energy flow management, blockchain security, advanced routing protocols, and energy storage. The papers [114,115,116,117] addressed resource allocation, error prediction, and security assurance techniques for smart grids use cases and the papers [118,119,120] presented different AL and ML models for future smart grid optimization, e.g., LSTM-MA for prediction accuracy, lightweight LSTM-RNN for load prediction, or deep reinforcement learning for dynamic resource allocation. In [121,122], a power meter for monitoring energy flow and helping with energy management was proposed. A survey from [123] addressed blockchain applications and highlighted recent research and practical projects. A power allocation optimization solution for maximizing energy efficiency in non-orthogonal multiple access systems was proposed in [124] and a testbed for Integrated Satellite-Terrestrial Network (ISTN) scenarios was studied in [125]. The paper [126] proposed a routing protocol for load-based route selection in 5G smart grids and [127] investigated the role of solid-state transformers and energy storage in smart grids.

#### 2.4.4. Resource Industries—Oil, Gas, and Mining

The energy and natural resources sectors are currently at a crucial turning point. These industries, which depend on skilled labor and intricate infrastructure, encounter distinct challenges for adapting to new requirements in the field. The use of new 5G/6G technologies can solve some of the problems with high-risk environments, workforce, or productivity and adaptability.

In the White Paper [128], Erikson mentions that the solution is connected workforce. This refers to the fact that the workforce can be optimized by interconnecting workers in real time. Using any type of device, connected from anywhere, greatly improves operability, thus bringing more productivity, and improving the quality of life of workers by providing the possibility to work remotely. This solution can be implemented using non-public networks (NPNs). NPNs are custom-designed wireless LANs (Local Area Networks) that offer greater reliability, high performance, security, and cost-efficiency compared to Wi-Fi or public cellular networks. These systems are crucial for the digital transformation of industries like mining, oil, and gas. They provide the reliable, secure, and flexible connectivity required to support the adoption, use, and growth of new technologies in these sectors [129]. Specifically for the oil and gas sector, private networks provide considerable advantages, including the capacity to connect remote locations and integrate legacy systems without requiring hardware replacement.

The main use cases on NPNs in mining, oil, and gas industry are the following [130]:Environmental monitoring of the air and water quality sensors and seismic activity detectors. Accurate data collection and examination enable environmental impact assessment and ensure regulatory compliance.Thermal cameras can identify equipment anomalies, and the insights gathered are transmitted over the network. This enables experts—who could be located anywhere globally—to diagnose problems in real-time. Advanced self-driving vehicles and automated robotic systems are capable of maneuvering through intricate settings, executing a diverse array of functions. These tasks span from conducting thorough inspections to managing materials with remarkable accuracy and productivity.Worker safety in remote and hazardous locations is critical. Wearable devices that feature biometric and GPS sensors can transmit, in real time, the health condition of workers and their exact location, allowing for close monitoring and rapid intervention in case of emergency.Supply chain optimization allows real-time tracking of shipments, equipment, and inventory, optimizing logistics, reducing delays, and lowering operational costs.Energy management systems enable precise control over power usage, reducing energy costs for sustainable operations.

The concept of private networks has been a topic of extensive discussion across various industry forums. The 5G 5G-ACIA in NPN White Paper [131] and the 3GPP in 5G Specification [132] made a categorization of NPN network types within the telecommunications and industrial sectors. According to this, there are several NPN topologies (see Figure 4).

The standalone NPN (SNPN) typically functions as an independent, self-sufficient system. This is a private network managed by an industrial operator that uses a privately licensed spectrum. Being a private network, access from public networks is restricted. Thus, SNPS uses dedicated 5G radio network components and owns storage solutions and a LAN to interconnect the devices used in the network. This architecture ensures a very high degree of security, speed, and efficiency. At the same time, not having access to the services and hardware components available in public networks, the upgrade process is slower and more expensive.

The second topology allows more operational flexibility by allowing SNPS access to the public network. Functions such as emergency voice and data services are essential. Additionally, features like shared RAN arrangements, backup or failover capabilities, and controlled access for external devices are also important components. In this topology, the control over the core network components remains under the control of the SNPS. Instead, spectrum provided by a Mobile Network Operator (MNO) is used. To ensure a high degree of security but also to provide additional functionalities or access to resources, access is allowed to external devices or services after successful authentication. The SNPS can also allow controlled access to public network services to devices based on needs.

In the case of the third topology, the integration of NPN with public networks is more complex. In this case, MNO and industry share a dedicated 5G RAN. Instead, network slicing is used to create private networks. This method is suitable for scenarios where low latency is important or data needs to be processed locally. The protection of the private network is achieved by implementing an access mechanism that isolates RAN cells from threats that may come from public networks.

The topologies presented above can provide solutions for any type of need such as flexibility, performance, resource allocation, or security.

Research studies in this area address issues such as Industry 4.0 transformation, MCX in underground mines, fleet management systems, 5G radio features in underground mines, 5G RAN solutions for the coal mining industry, and human safety management. The study [133] proposed a 5G-based virtualization-based solution for oil and gas fields, assuring high-band capability and low latency. The 5G New Radio Unlicensed spectrum is considered for realizing dense RAN. In a comparative analysis of LoRa and LoRaWAN protocols used in mine environments for mission-critical communications, Ref. [134] proposed a new protocol. This protocol aimed to reduce latency and enhance overall reliability. The paper [135] proposed a 5G architecture for underground fleet management systems that provide unified access to various sensors, video monitoring, and remote centralized control of trackless rubber tire vehicles and in [136] radio channel measurements for two 5G frequency bands in a real underground mine were performed. The paper [137] identified business scenarios, production applications, and industry needs for a coal mine. It proposed a 5G RAN solution and private network technology for solving the requirements of the coal industry. The studies [138] and [139] identified the key technologies used in 5G that can be successfully integrated in smart open-pit mines for providing the following services: unmanned driving, video streaming, unmanned driving systems monitoring, and remote control. In [140], an AI algorithm to optimize the transportation system for open-pit mines was proposed. 5G communication was used to collect a large amount of data from sensors. Trained with this dataset, the operation efficiency was improved and in [141], a light-based decision mechanism based on a human safety management algorithm to assess and identify conditions unsuitable for human life in mines was studied. To communicate emergency events in real-time, a Li-Fi channel was tested. To address issues related to low-power IoT communications used in resource industries, approaches in works [142] and [143] have proposed modulation techniques to achieve significantly higher spectral efficiency and data rates.

#### 2.4.5. Practical MCX Deployment Architectures in 4G/5G Tactical Environments

To provide a comprehensive perspective on practical and operationally validated MCX deployments, this subsection offers a high-level synthesis of tactical-grade solutions currently available on the market. These solutions exemplify technological maturity and deployment readiness, reinforcing the analytical findings and future-oriented conclusions outlined in this review.

Several advanced, 3GPP-compliant MCX platforms, formally authorized for deployment in Romania, have been identified. The Ericsson Tactical Bubble [144], Athonet HPE Tactical Cube [145], and Nokia Tactical Solutions [146] offer fully integrated Radio Access Network (RAN) and Core Network (CN) architectures, supporting robust, scalable, and mission-ready communication capabilities. These platforms are engineered for tactical, emergency, and mission-critical environments, and emphasize key operational attributes such as interoperability, secure service delivery, and flexible field deployment.

A comparative overview of the approved tactical MCX solutions available in Romania is provided in Table 4.

Each solution is tailored to specific deployment scenarios. Ericsson’s solution is ideal for secure integration with existing 5G infrastructure and is optimized for centralized control. Athonet HPE’s offering emphasizes modularity and flexibility via containerized network slicing, making it adaptable to both public safety and military domains. Nokia Tactical Solutions, particularly through the Nokia Perimeter platform, are designed for high-resilience, low-latency communications in resource-constrained or isolated environments, with a strong edge computing component.

MCX solutions are tightly integrated with 4G and/or 5G network infrastructures to ensure communication continuity during emergencies or in challenging operational contexts. Core services delivered by MCX platforms include MCPTT for real-time voice communications, MCData for secure transmission of essential data such as sensor information and multimedia content, and MCVideo for real-time high-definition video streaming to support situational awareness. These services are foundational in contexts where low latency, security, and reliability are mandatory.

To ensure end-to-end service integration across 4G/5G tactical networks, several MCX application suites with high interoperability and infrastructure transparency have also been evaluated as follows: Team on Mission (Streamwide) [147], Mission Critical Services (Leonardo) [148], and Nokia Team Comms [149].

In addition to infrastructure platforms, several standalone MCX application suites have been assessed for their interoperability, scalability, and operational relevance in 4G/5G tactical contexts. A comparative summary of these solutions is presented in Table 5, which highlights their communication features, security capabilities, and deployment performance across various mission-critical use cases.

Streamwide’s Team on Mission excels in scalability and integrated service delivery for mission teams. Leonardo’s MCX Services platform provides robust security and full interoperability for high-assurance operational contexts. Nokia’s Team Comms focuses on performance, leveraging edge computing to minimize latency and ensure dynamic adaptability.

These MCX applications are typically deployed as part of broader, fully integrated 4G/5G MCX solutions offered by the primary vendors. The integrated platforms evaluated in this context include Ericsson Tactical Bubble with Streamwide MCX, Athonet HPE Tactical Cube with Leonardo MCX, and Nokia Perimeter with Nokia Team Comms.

These end-to-end configurations represent practical, certified solutions for tactical and mission-critical deployments, demonstrating real-world viability and alignment with national security standards.

The analysis of authorized, 3GPP-compliant tactical MCX platforms and their interoperable application suites confirms that mission-critical communication solutions have reached a level of technological and operational maturity suitable for secure, scalable, and real-world deployment. Their formal approval for use in Romania highlights the national readiness to adopt standardized, high-assurance MCX capabilities in mission-critical infrastructures.

## 3. MCX Future Trends

Mission-critical services will evolve significantly over the next decade (2025–2035), seeking to leverage existing 5G technologies but also to advance to new heights of development through future 6G technologies. Future MCXs will feature exponentially higher data transfer rates, extremely low latency, and extensive connectivity capabilities. These advancements will leverage sophisticated automation, artificial intelligence, and IoT solutions to improve efficiency and reliability across sectors such as public safety, resource industries, healthcare, transportation, and utilities. These advances will enable resilient, self-organizing networks capable of autonomously managing emergency responses, thus ensuring an unprecedented level of security and seamless operation of vital services for society.

Currently, 5G is starting to enter the commercialization phase, and by 2030, the deployment of the next-generation network, referred to as 6G, is expected to take place. Consequently, this is an opportune moment to initiate studies on 6G. The future development of 6G communication networks must take into account ensuring global coverage. Up to now, terrestrial networks have predominantly been developed with iterative enhancements to their characteristics. These can no longer meet the growing needs of future applications. To address the challenges of communication scenarios occurring in the air or underwater, future designs must overcome current limitations. Additionally, they need to reduce the high costs associated with establishing dense cellular networks for global coverage. Therefore, upcoming systems will comprise a fully integrated communication network that spans space, air, land, and sea environments. This will involve using a combination of advanced technologies such as satellites for global communications, drones for rapid data delivery in remote areas, and dense terrestrial networks. In this way, a maximum coverage and capacity in urban areas will be provided [150].

Standardization efforts in this area are supported by several entities, which seek to provide both beneficiaries and manufacturers with an interoperable environment that will contribute to the rapid and efficient development of critical services. One of the most important groups aiming at MCX, the SA6 work group of 3GPP, is developing technical specifications that enable applications to operate efficiently and with a high degree of security in next-generation networks.

The ITU-R, through its reports, also provides clear guidance and perspectives into the technological and operational requirements needed to ensure global interoperability and effectiveness of critical communication systems in the 5G/6G context. The ITU’s proposed directions serve to establish a standardized framework for communication systems. They also offer essential recommendations for developing the infrastructure required to support critical communications on a global scale.

### 3.1. ITU-R Vision on Future Technologies Trends in the 6G Era

The radio communication sector of ITU has started the process of standardization of future 6G technologies by publishing the report ITU-R M.2516-0 [2]. This report offers an extensive overview of the future technical elements of terrestrial international mobile telecommunications systems (IMT), focusing on the period leading up to 2030 and beyond. It underscores significant emerging services, trends in applications, and relevant driving factors. The technologies discussed serve as a collection of potential enablers for future applications. Additionally, this report presents a toolkit of technological progress for terrestrial IMT systems, encompassing the evolution of IMT through technological improvements and their implementation. It also leaves open the possibility for the adoption of both existing and future emerging technologies. ITU has kept working in this direction with the publication of the recommendation ITU-R M.2160-0 [151], which defines the objectives and framework, as well as general goals, of the future development of IMT for 2030 and beyond. This report details the technological progress in terrestrial IMT systems, focusing on new emerging technologies that will be used in radio interfaces, mobile terminals, and radio access networks. IMT-2030 offers enhanced and new capabilities beyond those of IMT-2020. It supports expanded usage scenarios and enhanced key performance indicators. The capabilities proposed for IMT-2030 are detailed in Table 6.

Compared to IMT-2020, IMT 2030 includes the same usage scenarios but with improved features, as well as new ones based on new capabilities such as AI and sensing. The capabilities envisioned for IMT-2030 usage scenarios are [152], as follows:
Immersive Communication extends the capabilities of enhanced Mobile Broad Band (eMBB) by trying to provide users with interactive, rich, and immersive machine interfaces and video experience. By using a more efficient spectrum, services that are more consistent can be offered with higher transfer rates and mobility by providing communications in both urban and rural environments or in hard-to-reach areas. Typical use cases of immersive communication with applicability for MCX are the following: ultra-HD video streaming for natural disaster reconnaissance operations, telemedicine, and immersive remote multi-sensor systems for industrial automation and control or critical infrastructures monitoring;Hyper Reliable and Low-Latency Communication (HRLLC) extends the capabilities of URLLC to improve reliability and responsiveness. In this way, the communications will better support MCX for which these features are essential. Typical use cases of HRLLC for MCX are the following: full automation and control in industrial environments, in healthcare it can provide support for medical procedures where latency and responsiveness are essential, and real-time monitoring and control applications in smart grids or mining and gas industries;Massive Machine Communication extends the capabilities of massive machine-type communication (mMTC) beyond traditional machine-centered communication. Massive communication can be very useful in MCX applications where a number of low power consumption sensors are used. In addition, features such as high connection density and wide coverage can be very useful in providing special need services such as MCX;Ubiquitous Connectivity is a new usage scenario introduced in IMT 2030 that provides connectivity for anything, anytime, anywhere, even in areas not covered by existing communication networks. MCX can be used in applications such as Isolated Operation for Public Safety (IOPS), oil and gas platforms in isolated areas need permanent connectivity to public networks, or similar systems;Artificial Intelligence and Communication is also a new usage scenario introduced in IMT 2030. In future 6G networks, AI will improve every aspect of communication, such as decision-making, learning and reasoning. Typical MCX applications utilizing this usage scenario can be found in healthcare, in patient monitoring, smart cities, resource industry, smart grids, and autonomous vehicles, where AI can monitor and assist communications;Integrated Sensing and Communication, a new usage scenario in IMT 2030, enables high-accuracy sensing using communication signals. It provides information about connected and unconnected devices as well as the surrounding environment, offering a wide area of multi-dimensional sensing. MCX applications that can take advantage of this usage scenario are those involving surveillance, activity or intruder detection, navigation, environment detection (fire, flood, and hazardous gas emissions), and movement tracking. In order to provide these services, several key points must be fulfilled, as follows: accurate positioning, high-resolution sensors, and detection and mapping of persons or objects.

### 3.2. Future of Mission-Critical Services

#### 3.2.1. Mission-Critical Communication Enhancements Trends in 6G Era

MCX services are continuously developing. In Release 19, 3GPP, through its SA-6 working group, aims to standardize new functionalities. According to [73], the standardization efforts will focus on MCX and railway enhancements, the sharing of administrative configuration management information (MCShAC), and generic IOPS and MCX over NTNs.

Enhancements to MCX include the addition of a recording server that enables authorized logging, storage, and retrieval of communication metadata and media from private or group sessions. Another service will be introduced, namely mission-critical discreet listening, whereby authorized users will be able to follow the communications (MCPTT, MCData, and MCVideo) of a targeted group without their consent. Railway enhancements involve interfacing with GSM-R for MCPTT and MCData as well as the separation of media signaling into a mission-critical system. MCShAC will be enhanced by the fact that the following administrative configuration management information can be shared between interconnected mission-critical systems: group management configuration, and profile and system parameters. The Generic IOPS system using 5G will be able to benefit from a mobile system without backhaul. In addition, by using NTN, participants in communication groups will be able to be informed about the reduction of communication quality.

3GPP Release 19 was designed to unlock the full potential of 5G technology. It introduced new capabilities, made numerous improvements to existing 5G Advanced features, and established a path toward future 6G development. Achieving the full benefits of 5G also involved addressing real and urgent business needs. Additionally, it introduced new advanced capabilities included duplexing evolution, higher mid-band spectrum (i.e., 7–16 GHz), and integrated sensing and communication, generating new value in commercialization efforts as well as effectively enabling advanced deployments. Also, existing capabilities were enhanced, including further improving mobile experiences and extending 5G coverage into new areas and evolving new and improved devices and networks.

The future direction for 6G technology was added. This involves aligning with initial vision concepts, supporting fundamental research, and developing a timeline for the anticipated evolution of the technology. A detailed summary of the main features specified in Rel-19 is presented in Table 7.

#### 3.2.2. MCX Specific Requirements for 6G

The evolution of MCXs in the future 6G networks will take into account the use of certain technical enablers but will also require certain KPIs used to monitor their performance.

Systems that implement MCX must work under special conditions that satisfy different needs depending on the industry, organization, or specific service context. Thus, it is very difficult to establish and especially to implement KPIs that satisfy all the requirements imposed by MCX use cases. However, considering the general characteristics of these systems such as availability, reliability, performance, responsiveness, security, and operational efficiency, we will present below some of the KPIs that best reflect these aspects.

For MCX to perform effectively, it must be available and reliable. Among the most important KPIs that can measure these properties are Mean Time Between Failures (MTBF), Mean Time To Restore (MTTR), uptime percentage, and number of incidents. MTBF is the mean time between service interruptions. A higher MTBF indicates better reliability. MTTR is the average time taken to restore service after a failure. Lower MTTR values indicate faster restoration processes. Such systems must be available and fully functional 99% of the time. Uptime percentage is the KPI that measures this property. The number of incidents is a very good indicator to determine the reliability of a system. Performance and responsiveness are two other essential characteristics of systems implementing MCX. Among the most important KPIs that characterize these characteristics are response time for the service to respond to a user request or transaction, throughput—the number of transactions/queries handled per second, error rate—the percentage of failed requests or success rate—the percentage of successfully completed requests, and latency—the delay in data transfer or processing. For MCX, systems security is of utmost importance. The level of security can be characterized by several parameters. One of these is the Security Incident Frequency (SIF), which measures the number of detected breaches, intrusions, or vulnerabilities. Additionally, the Mean Time to Detect (MTTD) indicates the average time taken to identify a security threat. The Mean Time to Respond (MTTR) reflects the time needed to mitigate security incidents. Lastly, the compliance rate quantifies how well the organization adheres to relevant security standards and regulations.

Characteristically good operational efficiency can be achieved through high Service Layer Agreement (SLA) compliance. SLA compliance is important for MCXs because it directly impacts users’ perception of system reliability and performance. SLA compliance involves ensuring that the service meets the predefined performance standards and operational requirements agreed in the SLA. The SLAs define all the system KPIs mentioned earlier, as well as additional parameters related to Service Level Objectives (SLOs). These SLOs set specific performance targets for various aspects of the service, such as data processing speed, backup frequency, disaster recovery capabilities, and response times for support requests. They also include penalties or remedies if the service levels are not met. Implementing compliance SLA helps ensure the quality and reliability of the services, build trust with customers and stakeholders, identify areas for improvement, and maintain contractual obligations. This approach also helps avoid potential penalties for non-compliance.

Several technical enablers are anticipated to play a crucial role in 6G networks to support mission-critical services. Among the most important are the following: Advanced AI and Machine Learning, Non-Terrestrial Networks (NTN), Terahertz (THz) Communication, Advanced Network Slicing, Integrated Sensing and Communication (ISAC), Quantum Communication (QC), and edge computing. Since the first two have a major impact on the future development of MCX, they will be treated separately. For the other technical enablers, the main characteristics will be highlighted, with an emphasis on standardization efforts and major challenges.

THz Communication

Terahertz communications operate in the 0.1–10 THz range and are one of the technologies expected to be developed in future 6G networks. Offering latencies below 100 µs, distances of up to hundreds of meters, and ultra-high data transfer speeds in the order of hundreds of Gpbs, it is essential for the further development of MCX.

Standardization efforts have been initiated by the IEEE through the IEEE 802.15.3d standard. A detailed description of the standard, key design principles, target applications, and use cases is presented in [153]. ITU-T, through the ETSI report [154], identified frequency bands for THz communication systems, described the current regulatory situation, and identified services that may be considered for future studies. Implementing this new technology presents several technical challenges. Some of these are related to propagation and channel characteristics, such as severe path loss and molecular absorption, particularly caused by water vapor and oxygen, which restrict the effective communication range. Additionally, the technology has poor sensitivity to diffraction and blocking effects. There are also limitations due to the immature state of channel models for Terahertz (THz) frequencies, making accurate predictions and optimization more difficult. In terms of hardware component realization and integration with existing systems, the challenges are transceiver and antenna fabrication, which requires advanced technologies such as CMOS scaling, graphene, or photonic sources; beam alignment precision required by the extremely narrow beam width; and thermal and power constraints generated by high-frequency operation. Furthermore, interfacing THz communications with other 6G-specific technologies such as NTN, edge computing, AI-native, and advanced slicing architectures requires harmonized standards and interoperability frameworks.

Advanced Network Slicing

Using network slicing technology, the infrastructure of a physical network can be partitioned into multiple virtual or logical networks, called slices. In this way, each slice will be able to offer specific services to user groups. This technology offers numerous benefits. By automatically creating, modifying, or deleting slices in real time according to network needs, dynamic and autonomous management is made available. Given that slices can be managed across the entire network, end-to-end orchestration can be achieved. Precise control of resources enables fine-grained resource allocation. Security is achieved through the separation of slices. The use of AI/ML can enable slice performance optimization, resource needs prediction, and automated slice management tasks.

Standardization of the field was initiated by 3GPP in Release 15, which introduced the concept of network slicing and several basic elements of lifecycle management. Release 16 introduced SLA attributes and the concept of closed-loop automation. Release 17 introduces the technology into non-public network management and makes progress in terms of energy efficiency. Release 18 addresses issues such as exposing network segments to customers and intent-based network segment management. In Release 19, studies are being conducted to investigate the possibility of interfacing with external consumers. The technical specifications produced by 3GPP on this subject can be found in [155].

Implementing these concepts in existing or future solutions faces several challenges. Real-time management requires predictive systems that can dynamically adjust slices using artificial intelligence models. This introduces complexity in terms of computation and integration.

To reduce interference between slices and allocate resources efficiently, advanced resource management algorithms and dynamic slice adaptation mechanisms are required. Integrating network slicing into existing network infrastructure can be difficult, especially when working with legacy equipment. To achieve the performances required by the new 6G networks, a large number of slices need to be integrated. This must be supported by the system architecture. For MCX in particular, traffic prioritization is necessary, which means that techniques such as resource perforation are required. Those techniques must be optimized for operation in real-world conditions.

Integrated Sensing and Communication

The 6G communications of the future will consider a closer interconnection between communication and sensing. ISAC is one of the emerging technologies that integrates sensing and communication systems through shared hardware, signal processing algorithms, and spectrum resources. In this way, ISAC offers significant potential for improving mission-critical services characterized by strict reliability, latency, and accuracy requirements. By enabling joint waveform design, hardware reuse, and dynamic spectrum sharing, ISAC improves spectrum efficiency, provides ultra-low latency, and improves reliability, coverage, and situational awareness. However, there are some challenges in implementing these solutions. Simultaneous communication and sensing operations can lead to interference between the two functionalities and resource allocation issues. The complexity of signal processing will increase power consumption. In some use cases, hardware limitations cannot guarantee these additional processing capabilities.

3GPP, through the activities of Service and System Aspects Work Group 1 (SA WG1), identifies potential use cases and requirements for implementing ISAC services in 5G systems [156]. ITU-T, through its recommendations for International Mobile Telecommunication 2030, lays the foundation for the fusion of IoT sensors and communications [157].

Integrated Quantum Communication (IQC)

Integrated Quantum Communication is one of the promising technologies that can revolutionize 6G communications. In the context of MCX, it can improve the security and reliability of communications by providing new capabilities such as secure key distribution, quantum sensing, and distributed quantum computing. This technology promises unbreakable security through the Quantum Key Distribution (QKD) mechanism, guaranteeing the protection of cryptographic keys by the laws of physics, regardless of the computational power of the eavesdropper. QKD protocols also allow the communicating parties to detect the presence of any eavesdropper attempting to intercept the quantum signal. Given that quantum computers are becoming increasingly powerful and stable, classical cryptography may be threatened in the near future. IQC may be a future solution to replace it. However, given that this technology is still in its early stages, there are still many issues to be solved for its implementation and use. Currently, there are still limitations on the distance over which quantum signals can be transmitted. This is because quantum signals are vulnerable to loss and decoherence as they travel through optical fibers, which restricts the effective communication range. Another unresolved issue is qubit coherence, as maintaining quantum states in noisy environments remains a challenge. Ensuring communication for a large number of users is a necessity for MCX. For now, scaling quantum communication networks to support a large number of users is a significant challenge. In addition, there are relatively high costs and issues related to integration with existing communication networks.

Organizations in the field have begun the standardization process by conducting studies or defining the main infrastructures, functionalities, and interfaces. ITU-T, through publication Y.3802 [158], establishes the functional architecture of QKD networks. ETSI, through its publications GS QKD 004 [159] and GS QKD 014 [160], specifies QKD application interfaces and the QKD protocol and data format of the REST-based key delivery API.

Edge computing

Edge Computing is a pivotal technical enabler for 6G networks, and is also essential for specific services such as MCX, ensuring performance characteristics such as ultra-low latency, high reliability, and enhanced security. The main idea behind this technology is to decentralize computing and storage to the edge of the network, close to end users and data sources. Edge computing is an alternative to cloud-based architectures that struggle to meet performance indicators such as latency and bandwidth.

Edge computing offers several advantages that are particularly relevant to MCX. By processing data closer to the source, edge computing significantly reduces network latency. High reliability is ensured by providing redundant computing resources at the edge of the network. Data security and privacy can be achieved by processing data locally, without the need to transmit it over unsecured channels. By reducing the amount of data that needs to be transmitted over the network, bandwidth optimization is achieved. Scalability can be improved by adding new edge servers, allowing network operators to more easily scale their computing resources.

The standardization process for integrating this technology into 5G networks is at an advanced stage. 3GPP began introducing edge computing as early as Release 15, but it was not until release 17 that specific improvements were offered. Release 18 brings improvements in roaming to support access to the edge home environment in a public terrestrial mobile network, and Release 19 brings further improvements to the edge application server [161]. ITU-T has also addressed this area. X.1648 specifies guidelines for edge computing data security, Y.4122 details the requirements and capability framework for an edge computing-compatible gateway in IoT, and ITU-T Y.FMC-EC addresses unified edge computing to support fixed-mobile convergence in IMT-2020 networks.

#### 3.2.3. Non-Terrestrial Networks

Non-Terrestrial Networks (NTN) can have a major impact on the future development of MCX by providing reliable and resilient communications. Infrastructures relying on NTNs at the disposal of satellite systems and aerial platforms can provide good quality connectivity in challenging environments where terrestrial networks may be unreliable or non-existent. These networks, which include Low Earth Orbit (LEO), Medium Earth Orbit (MEO), Geostationary Earth Orbit (GEO), and High Altitude Platform Stations (HAPS) satellite communications such as airplanes, balloons, airships, and UAVs, provide distinctive capabilities that can complement and enhance terrestrial networks in various mission-critical scenarios [162].

Recognizing the demonstrated advantages of NTNs in enhancing resilience and expanding coverage for mission-critical services, 3GPP has initiated efforts to adapt communication protocols to satellite-specific requirements. These efforts have materialized in the modification of 5G NR, narrowband IoT, and machine-type communications for interconnection with satellite networks. According to the Ericsson report [163], efforts are being directed at integrating satellite technologies with standard cellular communications to improve connectivity parameters and broaden coverage area.

3GPP in Releases 17 and 18 started to define NTN architectures for satellite communications. The first approach is based on a transparent (bent-pipe) architecture.

In this approach, a transparent payload is used. The satellite acts as a repeater in this setup. This is because the RAN node of the 5GNR network, known as the gNB, is positioned on the ground behind the gateway, and the satellite helps facilitate communication between these components. The role of the satellite is limited to basic radio frequency processing such as frequency conversion, amplification, and beam management. Since Release 19, the architecture has changed in that the gNB node is placed on the satellite. This architecture with regenerative payload is more flexible as it allows packet switching directly within the payload using inter-satellite links. It also offers better resilience as it is able to route traffic and deliver calls directly between satellites. The higher performance is provided due to significantly lower latency for procedures between the gNB and user equipment.

In 5G networks, Non-Terrestrial Networks (NTNs) are integrated separately as needed. However, in 6G networks, it is planned to achieve full integration of NTNs, allowing access to non-terrestrial components to be inherently part of the network. This will be realized through a comprehensive 3D, multi-dimensional, multi-layer, and multi-band architecture [164]. This vision, embraced by both mobile and satellite network operators, is being integrated within the 3GPP standardization efforts. This integration will require new types of interfaces that offer flexibility and reconfigurability to provide interoperability between different node and channel characteristics, and different business models.

The NTN standardization plans of 3GPP and ITU-R will start with Release 20. 3GPP and has triggered the study phase, intending to move to the normative phase by 2027 and Release 21. At the same time, ITU-R intends to standardize NTN by the end of 2029.

In order to concretize the attempts to integrate the two types of networks, we have summarized the key characteristics across network generations in Table 8.

Even if the benefits that the implementation of NTNs in future MCX architectures will bring are obvious and steps have already been taken to standardize this area, there are still issues that require research efforts to identify solutions applicable in practice. Among the most frequent and important practical challenges encountered in NTN implementation are spectrum coexistence, rain fade, regulatory constraints, and terminal power consumption.

NTNs in most cases share the frequency spectrum with terrestrial devices or other NTNs, thus creating potential interference problems. For example, HAPS and LEO typically operate in the Ku-, Ka-, and Q/V bands, potentially interfering with existing terrestrial or satellite systems or other NTNs. These problems can be aggravated in crowded spectrum regions, such as urban areas, where severe co-channel and inter-cell interference can occur. One of the identified solutions is dynamic spectrum sharing. This approach enables real-time management of the available frequencies by allowing their flexible and efficient utilization at any given moment [165]. Also, optimizing interference coordination using centralized, distributed, or combined solutions [166] can have remarkable effects. Due to the use of high-frequency bands, NTNs may have problems with signal attenuation in unsuitable atmospheric conditions. Atmospheric moisture can negatively influence communications through absorption and scattering of radio signals. This is much more frequent and intense in use cases suitable for MCX communications in tropical or heavy-precipitation regions that are often disaster-prone and underserved. LEO satellites can be affected by fading and scintillation due to varying elevation angles and beam sweep. HAPS platforms, even if operating above the weather layer, can be affected by rain fading during feeder-to-ground transfers. Among the most widely used solutions to solve these problems are link diversity [167], adaptive coding [168], and traffic rerouting algorithms [169].

All these solutions mentioned above will require additional processing for the satellite platforms, which will result in additional power consumption, which cannot be easily achieved on such platforms. The currently proposed solutions for power consumption efficiency would be miniaturized and highly efficient antennas, and using deep reinforcement learning for adaptive offloading of signal processing tasks to conserve terminal power based on real-time channel, traffic, and platform [170], or energy harvesting solutions.

Regulatory frameworks are still fragmented across different countries and international borders. Since NTNs, such as LEO satellite territories and HAPS platforms, are used across borders, issues related to cross-border licensing, liability, and spectrum rights often arise. As a result, the HAPS platform must comply with regulations from both telecom and aviation authorities, making the management of these regulatory challenges quite complex. Some efforts in this directive have started. The Standardized European Rules of the Air and the Air Traffic Management/Air Navigation Services Regulation are essential parts of the Single European Sky initiative. Their goal is to harmonize various aspects of air traffic management, including communications, navigation, and surveillance. This harmonization also includes the integration of new technologies such as drones and electric vertical take-off and landing aircraft. ITU-T through the World Radiocommunication Conference 2023 has established general requirements for the radio frequency spectrum and satellite orbits.

Attempts to integrate NTNs into existing terrestrial communication networks are at an early stage. Some research efforts in this area have been concretized in articles and book chapters. Preliminary research directions include the following aspects: identification and analysis of critical service requirements for NTN, NTN offloading and evaluation, analysis of possible 6G NTN architectures, examination of the motivation for NTN integration, and challenges for large-scale IoT deployments. In [171], integration issues between NT and NTN systems that provide MCX were detailed. The paper considered two architectures based on the proposals made for 3GPP Release 20. Also, a solution for seamless Vertical Handover between TN and NTN was proposed in the second part of the paper. The study was performed in the EU 5G-GOVSATCOM project. The study [172] analyzed the computationally intensive eXtended Reality applications used in MCX that can be offloaded in the integrated TN/NTN environment. The study [173] explained the need for deterministic networking capabilities of the current network in the context of time-sensitive, ultra-reliable holographic mission-critical applications. It also proposed a deep reinforcement learning-based deterministic network selection and routing scheme to manage end-to-end delays in holographic services. In [174,175], the evolution, standardization architecture, research challenges for integration NTN in 5G/6G, and the benefits resulting from this integration were summarized.

#### 3.2.4. AI/ML Improving the Future of MCX

The integration of AI into mission-critical services introduces transformative capabilities, from intelligent automation to proactive threat detection and response. The methods by which AI can be involved in the further development of MCX are varied. AI can be used in predictive maintenance by identifying the optimal time to perform it. Repetitive automation processes can be taken over by AI, giving operators the time to solve more complex tasks. Real-time analytics can improve decision-making. Being implemented in cyber defense systems, AI components can identify and protect MCX against cyber attacks. Last but not least, AI systems can monitor and operate in hazardous conditions, ensuring the safety and effectiveness of human operators working in such conditions.

Although AI is actively explored for its capacity to improve operational efficiency, responsiveness, and security in communication networks, its integration into mission-critical services still raises important concerns. In [176], a number of challenges were identified regarding the use of AI in MCX: model bias, adversarial attacks, data security, model security, trust, explainability, and self-assessment. These problems can be addressed by strictly controlling the data and methods for training AI models, ensuring the security of the data and models used, and using concepts such as explainable AI (XAI) [177,178] to increase the trust in AI decisions.

The introduction of AI in MCX systems can also influence other performance parameters such as signaling overhead, model-training latency, or terminal compute requirements. Communication overhead including signalization processes can be influenced by AI in both positive and negative ways. For example, AI can be used to predict user movement behavior (movement speed and direction) so that cell selection and switching can be performed in advance to reduce signaling overhead [179]. Using AI together with edge computing techniques allows for process optimization, data filtering, decision-making, facilitating fast response and decreasing the amount of data transmitted [180]. Also, AI lifecycle management protocols, introduced by 3GPP starting with Release 16, add signaling interfaces between base stations and user equipment [181]. In contrast, techniques such as service-based RAN orchestration [182] and channel state information compression/prediction can reduce the negative effects introduced by AI signaling overhead. MCX use cases require rapid adaptation to changing environments. The training speed of AI models may reduce responsiveness or even compromise reliability during emergencies. This problem has been addressed using different approaches. Centralized or cloud-based training leverage powerful computational resources. This approach offers high accuracy and scalability, while also reducing the latency involved in model training. A federated learning and edge-based training approach offers the possibility to train models locally by sending only their parameters to a central aggregator. Even not sending the raw models still introduces latency overhead in the communication channels. To reduce latency overhead, several solutions have been proposed. These include cooperative federated learning, which uses edge server clusters to pre-aggregate data before global synchronization, as discussed in [183]. Additionally, techniques like gradient compression and dynamic sampling are used to decrease the amount of data transmitted during each training round, as described in [184]. As previously mentioned, some of the problems can be solved by using edge computing for AI needs which means that terminal compute requirements will grow. Since many of the MCX terminals do not have unlimited processing capabilities, several techniques have been proposed for this purpose. One approach [185] is the use of Neural Processing Units (NPUs), chips specifically designed for fast and efficient processing aiming to optimize AI performance with minimal energy usage. Software-level optimizations are another approach. Techniques like model pruning, quantization, and compact model architectures like TinyML reduce compute and memory footprint for constrained devices [186].

AI integration in MCX can be realized using the concept of native AI as a service. According to [187], there are several approaches for introducing AI into communication systems. The first approach involves replacing existing functionalities with AI techniques. The second approach involves adding new AI components to the existing architecture. These two approaches cannot always ensure backward compatibility. Thus, the third approach attempts to ensure alignment with existing interfaces by integrating AI-based components. This will provide control for existing components. However, all these three approaches cannot be defined as AI-native. An AI-native implementation includes AI-aware control elements capable of learning and reasoning. It also incorporates tools for managing the full lifecycle of AI-based modules. Figure 5 visualizes all these concepts.

The integration of AI in future communication systems is one of the standardization objectives of 3GPP. Priorities considered in 3GPP groups for AI/ML integration include [188] infrastructure and operator control, performance monitoring, AI based air interface extension, AI/ML model activation and deactivation, data collection, and AI/ML model transfer and delivery. Another standardization body dealing with this domain is the RAN WG2 which develops specifications for AI/ML Framework and Management Functions. They establish the workflow for managing AI/ML models, including stages such as capability discovery, training and selection, deployment and inference, performance monitoring, retraining, and eventual termination. Additionally, the ITU Focus Group on Machine Learning for Future Networks, including 5G, has developed a suite of standards. These standards address various aspects such as ML use cases, architectures, orchestration, data management, and marketplaces within IMT 2020 and 5G networks. Table 9 shows the main standardization efforts undertaken by these three bodies.

According to the O-RAN WG2 report [209], AI/ML algorithms can be used for Quality of Experience (QoE) optimization and traffic steering uses cases. Supervised learning algorithms such as Convolutional Neural Networks (CNN) and Deep Neural Networks (DNN) are used for service type classification. Long Short-Term Memory (LSTM) and XGBoost are employed to predict network Key Quality Indicators (KQI) and QoE, and for available radio bandwidth prediction, DNN algorithms are used. For traffic steering uses cases, Support Vector Regression (SVR) is used. DNN algorithms are proposed for cell load and user traffic volume prediction. Radio fingerprint can be predicted using SVR and Gradient Boosted Decision Trees (GBDTs).

The ITU-T report [208], which addresses the use of AI/ML in IMT-2020 use cases, specifies the types of algorithms that can be used. Thus, supervised learning algorithms such as CNN, DNN, LSTM, SVR, and GBDT can be used for traffic classification and prediction of resource requirements for network slices. Unsupervised learning techniques are mentioned for tasks such as deciding appropriate resource levels for different types of services in network slicing. Techniques for maximizing the utility of sliced backhaul and model optimization use reinforcement learning. Also, 3D-CNNs and convolutional LSTMs solutions are proposed for accurate long-term forecasting of mobile traffic throughout the network.

3GPP approaches differ from O-RAN and ITU-T. The vendor’s efforts focus on standardizing the infrastructure around AI/ML, not the algorithm models themselves [215]. Areas where AI/ML algorithms can be used include operator control, data security, and facilitating data collection and model management.

In parallel with standardization efforts, the academic community proposes different solutions for introducing AI in MCX applications. The key aspects that can be improved through AI integration that have been addressed in the research activities are network slicing optimization, mission-critical Internet of Things, overload detection and prediction, URLLC in 5G/6G, RAN enhancements, and UAV and Voice Assistant AI powered systems. Network slicing optimization solution is proposed in paper [216] that provided a solution for optimizing end-to-end network slicing using AI native implemented at the core network. The paper [217] proposed an advanced 5G/6G communication platform based on multi access computing, and AI and network slicing for mission-critical applications in smart cities. The paper [218] demonstrated proof-of-concept approaches for 5G network slicing in mission-critical use cases. In [5], an AI-based algorithm for mission-critical IoT applications to improve the age of information metric used to capture and evaluate information freshness at the destination was proposed. A solution for a proactive overload detection based on Machine Learning in a typical 5G core network was proposed in [100] and ML algorithm for overload prediction in [111]. A voice command virtual assistant powered by AI for mission-critical crises was described [219] and an AI-empowered UAV system search and rescue operations system which provide sensors, image recognition and autonomous navigation capabilities in [220]. Also, an analysis of the using of ML solutions for assuring 6G URLLC service and a case study where the problems of a URLLC channel were solved using centralized and federated deep reinforcement learning were performed in [221]. The paper [222] proposed an O-RAN architecture that supports split-plane multi-component cooperative AI models that utilize multiple centric RAN intelligent controllers and centric multi-access edge computing control loop.

## 4. Conclusions

Mission-critical services have rapidly evolved from narrowband analogue communications to standardized broadband systems, now supporting diverse services across multiple domains. This review highlights the evolution of MCX by examining standardization efforts of key organizations such as 3GPP, ETSI, the O-RAN Alliance, GCF, and BroadEU.net, alongside relevant research and development activities.

Current developments in MCX were analyzed, focusing on mission-critical communication aspects, such as MCPTT, MCData, MCVideo, service frameworks, vertical enablers, and mission-critical verticals. For each service type, essential characteristics, historical evolution, and current research trends were identified and summarized. Major mission-critical verticals, including public safety, emergency services, transportation, power grids, and resource industries, were analyzed from evolutionary, architectural, and functional perspectives. The analysis also highlighted existing challenges and emerging scientific developments.

The significance of service frameworks and vertical enablers such as the Common API Framework Service Enabler Architecture Layer, and Edge Application Enablement, developed within 3GPP, was underscored. These frameworks significantly facilitate the rapid and efficient implementation and integration of vertical-specific applications, thus contributing to performance optimization and accelerated adoption of new technologies in both current 5G and future 6G networks.

Detailed analyses of critical verticals revealed distinct needs and unique challenges across public safety, transportation (Future Railway Mobile Communication System), power grids, and resource industries. This underscores the necessity of tailored approaches for MCX integration, involving adapting standards and infrastructure to meet industry-specific requirements, ensuring interoperability, and achieving the required performance in complex and varied operational scenarios.

Security and resilience emerged as critical factors requiring significant attention. Integrating MCX into public and private networks demands advanced cybersecurity solutions and trust mechanisms. This requirement is further emphasized given the increasing use of Artificial Intelligence and Non-Terrestrial Networks, which introduce additional vulnerabilities and complexities related to sensitive data protection.

Artificial Intelligence and automation are identified as essential drivers of future MCX advancements, aiding network management improvements, resource optimization, and incident prediction. Despite their promise, these technologies require further research to address transparency, trustworthiness, and control over automated decision-making processes.

The integration of Non-Terrestrial Networks represents a substantial advancement, promising to extend coverage and enhance the resilience of critical communications significantly. Nevertheless, practical and integrated NTN implementation within MCX networks requires extensive ongoing research and standardization efforts, still being in the early stages of technical and commercial development.

In addition to future-oriented developments, it is important to note the set of frozen features introduced in 3GPP Release 19 that strengthen MCX capabilities in operational 5G networks. These include enhancements to MCPTT, MCData, and MCVideo aimed at reducing latency and improving reliability in mission-critical scenarios; native integration with Non-Terrestrial Networks to extend coverage in underserved areas; support for Reduced Capability devices optimized for cost and power efficiency; advanced positioning services enabling sub-meter accuracy for applications such as rescue operations and field coordination; extended network slicing capabilities tailored to Public Protection and Disaster Relief (PPDR) services; and reinforced security mechanisms, including improved cryptographic key management, greater resilience of Over-the-Air Rekeying processes, and the integration of Zero Trust principles. Once finalized, these specifications provide a solid foundation for scalable and interoperable implementations that meet both current and future MCX requirements.

Future trends in MCX development align closely with the ITU-R vision for 6G. This review identified proposed capabilities and key performance indicators for IMT-2030, explicitly connecting them to future MCX usage scenarios. Two key technologies—Non-Terrestrial Networks and Artificial Intelligence—emerge as critical enablers, significantly enhancing reliability, resilience, efficiency, and agility. These technologies were evaluated considering their benefits, integration challenges, and ongoing standardization efforts.

Mission-critical services are currently undergoing significant transformation, primarily driven by integration within 5G networks. Although most technical specifications have been standardized, further improvements and large-scale deployment efforts continue. The availability of fully integrated, authorized MCX solutions—such as those offered by Ericsson, Athonet HPE, and Nokia—confirms that current technologies are capable of supporting scalable, secure, and standards-based deployments for mission-critical services. Their operational validation in Romania reflects a broader trend of convergence between research, standardization, and field implementation.

Future development of MCX is closely linked to the emergence of 6G networks, where specific use cases are currently being identified to meet market demands. Ultimately, the introduction of 6G capabilities is expected to significantly enhance MCX functionalities, addressing increasingly specialized user requirements.

## Figures and Tables

**Figure 1 sensors-25-05156-f001:**
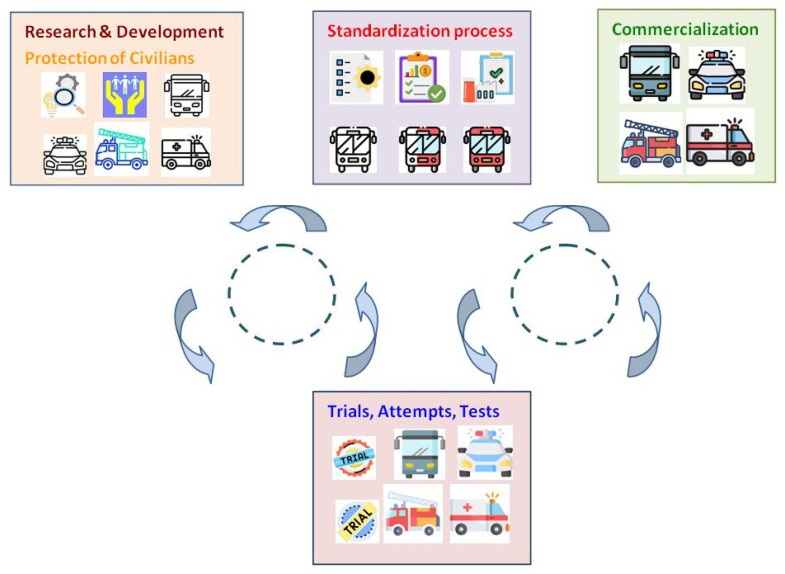
GPP in the standardization process.

**Figure 2 sensors-25-05156-f002:**
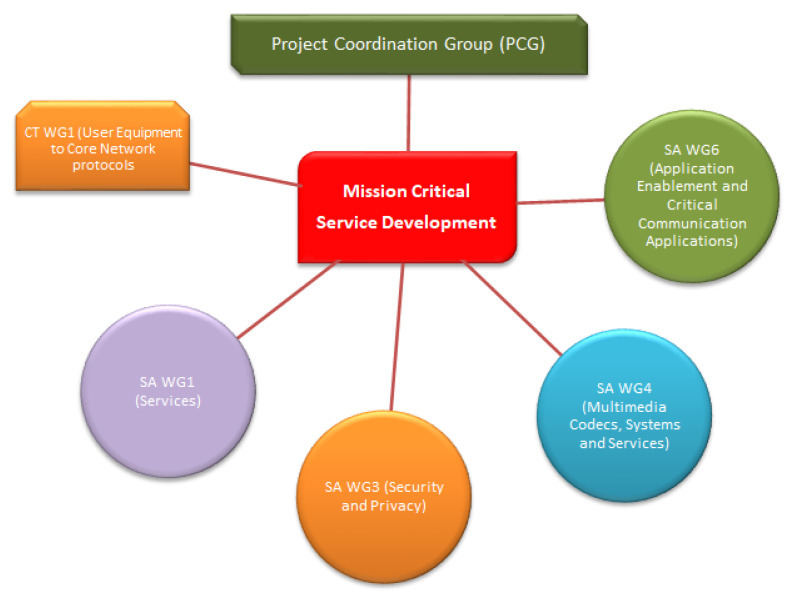
3GPP work on mission critical services.

**Figure 3 sensors-25-05156-f003:**
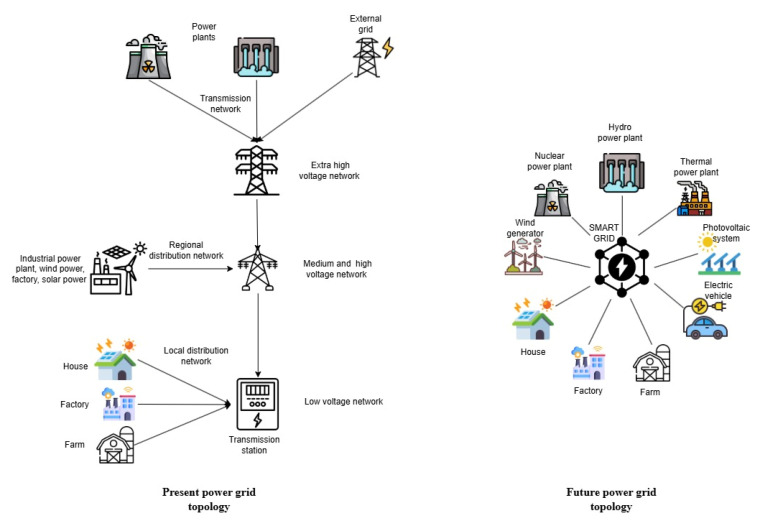
Typical topology of a power grid (present and future).

**Figure 4 sensors-25-05156-f004:**
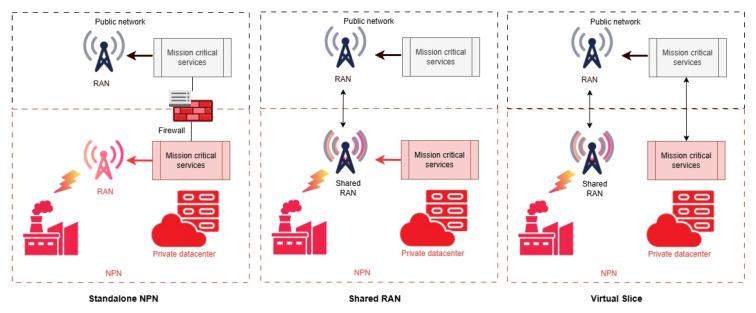
NPN—public network topologies.

**Figure 5 sensors-25-05156-f005:**
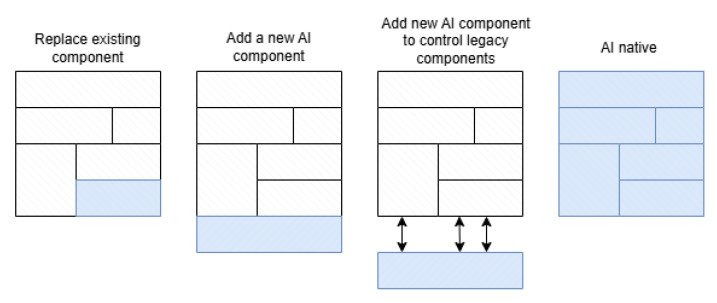
AI integration in existing systems.

**Table 1 sensors-25-05156-t001:** MCX standardization effort.

Technology	Release	Mission-Critical Services	Service Frameworks	Vertical Enablers
LTE Advanced Pro	Rel-13	MCPTT		
	Rel-14	MCPTT 2.0 MCVideo MCData		
5G	Rel-15	MCPTT 3.0 MCVideo 2.0 MCData 2.0 MC Interworking Railways	CAPIF	
	Rel-16	MCPTT 4.0 MCData 3.0 MC Interworking 2.0 Railways 2.0 MC MBMS API MC Location (Study) MC Logging	CAPIF 2.0 SEAL	V2XAPP
	Rel-17	MCPTT 5.0 MCData 4.0 Railways 3.0 MCIOPS MCOver5GS	SEAL 2.0 Edge Applications (EDGEAPP)	V2XAPP 2.0 5G Message Service (5GMARCH) Factories of the Future (FFAPP Study) Uncrewed Aerial Systems (UASAPP)
5G Advanced	Rel-18	MCX 6.0 MGWUE Railways 4.0 MCOver5MBS MCOver5GproSe MCAHGC MCShAC (Study)	EDGEAPP 2.0 SEAL 3.0 SEALDD ADAES NSCALE 5GFLS SNAAPP	PINAPP V2XAPP 3.0 UASAPP 2.0 5GMARCH 2.0
	Rel-19	MCX 7.0 Railways 5.0 MCShAC	EDGEAPP 3.0 SEALDD 2.0 CAPIF 3.0 CAPIF_EXT AIMLAPP, eLSApp Metaverse_App XRM_Ph2_App	5GMARCH 3.0 UASAPP 3.0 MMTel_App 5GSAT_Ph3_APP

**Table 2 sensors-25-05156-t002:** The evolution of 4G/5G technologies.

Technology	Release	Key Characteristics
LTE (Long-Term Evolution)	Rel-8/Rel-9	**(Also known as 3.95G or 4G LTE)** Foundation: The initial 4G technology standard. **Main features, according to ETSI TR 125 913 V8.0.0 (2009-01)**: [15]: Peak data rates: e.g., 100 Mbps (downlink) and 50 Mbps (uplink). Reduced latency: below 10 ms (possibility for a radio-access network latency), less than 100 ms (possibility to exchange user-plane data starting from camped-state with a transition time). Peak spectrum efficiency: improved, e.g., 2–4 × Release 6. Increased bitrates at the edge of cells, whilst maintaining same site locations as deployed today. Scalable bandwidth (for greater flexibility in frequency allocations), support for inter-working with existing 3G systems and non-3GPP specified systems, and further enhanced MBMS. Reduced capital (CAPEX) and operational expenditure (OPEX) including backhaul. Reasonable system and terminal complexity, cost, and power consumption. Support for inter-working with existing 3G systems and non-3GPP specified systems, and backwards compatibility. Efficient support of the various types of services: especially from the PS domain, e.g., Voice over IP, Presence. Optimized for low mobile speed but supporting high mobile speed: up to 500 km/h. Applications: Voice, Mobile TV, and Mobile Internet. Relevant documentation for LTE [16,17]: 3GPP TR 25.912 [18], 3GPP TR 25.913 [19], and 3GPP TR 23.882 [20].
LTE Advanced	Rel-10/Rel-11/Rel-12	**(Also known as LTE+, LTE-A, or 4G+)** Enhancement: An evolution and improved version of LTE, introducing features like carrier aggregation, which allows combining multiple frequency bands to increase bandwidth (data speeds) and higher-order modulation and MIMO technologies. **Main features, according to** [21]: Peak data rates: significantly increased over LTE; 100 Mbit/s for high and 1 Gbit/s for low mobility were established as targets for research. Latency: lower latency than LTE, less than 50 ms (from Idle mode (with IP address allocated) to Connected mode), less than 10 ms (from a “dormant state” in Connected Mode). Peak spectrum efficiency: supports downlink peak spectrum efficiency of 30 bps/Hz and uplink peak spectrum efficiency of 15 bps/Hz. A high degree of commonality of functionality worldwide, combined with the flexibility to support a wide range of services and applications in a cost-effective manner. Service compatibility across International Mobile Telecommunications (IMT) networks and fixed networks. Interoperability with other radio access systems. User equipment suitable for worldwide use. User experience: easy-to-use applications, services and equipment, high quality mobile services, improved performance at cell edges and increased number of active subscribers, and World Wide Web roaming capability. Relevant documentation for LTE-Advanced [16,17]: 3GPP TR 36.912 [22], 3GPP TR 36.913 [23], RP-080763 [24], and RP-090939 [25].
LTE Advanced Pro	Rel-13/Rel-14	**(Also known as LTE-A Pro, 4.5G, 4.5G Pro, 4.9G, or Pre-5G)** Further Improvement: Builds upon LTE Advanced with further enhancements to carrier aggregation and other technologies. **Main features, according to** [26]: Peak data rates: significantly higher peak data rates than LTE Advanced, often approaching Gigabit speeds. Speeds: significantly higher peak data rates than LTE Advanced, often approaching Gigabit speeds (providing peak speeds up to 2 Gbps, utilizing both licensed and unlicensed spectrum bands. It offers peak data rates exceeding 3 Gbps and introduces technologies like higher-order MIMO and 256QAM for increased capacity and performance). Latency: lower latency than LTE Advanced, less than 2 ms. LTE IoT: connects and manages many IoT devices, years of battery life, extended coverage. Private LTE: secure, high-performance cellular technology to private and local networks, custom-built and managed for specific uses. Cellular V2X: vehicle-to-vehicle connectivity, vehicle-to-infrastructure connectivity. Terrestrial broadcast: uses mobile networks to deliver public broadcast services (such as terrestrial live TV). Transition to 5G: often seen as a stepping stone towards 5G deployment. The parallel evolution of LTE Advanced Pro and 5G NR includes massive MIMO, beamforming and tracking for millimeter wave bands, scalable OFDM and flexible frame structure for wireless links, and advanced channel coding. Relevant documentation for LTE-Advanced PRO [16,17]: 3GPP TS 36.-series specifications (if only LTE is affected) [27] or 3GPP TS 37.-series specifications (if also other radio access technologies like UMTS or GERAN or NR are covered in this specification) [28], 3GPP TS 21.201 [29], and 3GPP TS 23.003 [30].
5G	Rel-15/Rel-16/Rel-17	**(Fifth Generation)** The fifth-generation mobile network increased capacity compared to LTE. It also introduces new features like network slicing and improved energy efficiency. **Main features, according to** [31]: Peak data rates: significantly faster than LTE, with peak speeds exceeding it, e.g., 20 Gbps (downlink) and 10 Gbps (uplink). Latency: ultra-low latency, as low as 1 ms (URLLC User Plane). New Radio (NR): utilizes a new radio technology (5G NR) with high-frequency radio waves. Capacity: handles a far greater number of devices than LTE. Use Cases: enables applications like autonomous vehicles, IoT, and real-time applications. Applications: wearable devices, IoT, Smart Cities. Relevant documentation for 5G [32]: 3GPP TR 38.900 [33], 3GPP TR 38.901 [34], 3GPP TR 38.913 [35], ITU-R M.2083 [36], ITU-R M.2410 [37], 3GPP TR 38.912 [38], 3GPP TR 38.801 [39], 3GPP TR 37.910 [40], and 3GPP TS 22.261 [31].
5G Advanced	Rel-18/Rel-19	**(Also known as 5.5G or 5G-A)** Refinement: refines existing 5G technology, a set of enhancements to the 5G standard, focusing on even higher data rates, lower latency, and expanded capabilities for new use cases (massive connectivity, higher capacity, and network slicing), improved energy efficiency, and expanded spectrum. Future-proofing: prepares for future technologies and applications that will be built upon 5G. **Main features, according to** [41]: Strengthens the end-to-end 5G system foundation (Advanced DL/UL MIMO; Mobile IAB, smart repeater; and AI/ML data-driven, enhanced mobility, evolved duplexing, and green networks). Proliferates 5G to virtually all devices and use cases (boundless extended reality; expanded sidelink; drones and expanded satellites communications; NR-Light (RedCap) evolution; multicast and other enhancements). Documentation for 5G-Advanced [32]: RP-233981 [42], 3GPP TR 37.911 [43], 3GPP TR 21.205 [44], 3GPP TS 38 series specifications [45], 3GPP TS 37 series specifications [28], 3GPP TS 36 series specifications [27], 3GPP TS 23.501 [46], 3GPP TS 23.502 [47], 3GPP TS 38.300 [48], 3GPP TS 37.340 [49], and 3GPP TS 38.401 [50].

**Table 3 sensors-25-05156-t003:** PPDR KPI metrics.

KPI Category	KPI Metric	Target Value
Reliability and Availability	Availability (Uptime)	99.99% (Four Nines)
	Grade of Service	<500 ms call setup success rate at peak load.
	Redundancy and Backup Systems	N+1 redundancy for core network components.
Coverage and Capacity	Geographic Coverage Area	99% of designated region (national, state, local).
	Indoor Coverage	65% signal strength inside buildings.
	Cell Throughput	≥256 kbit/s per user during peak.
Interoperability	Adherence to Common Standards	Compliance with applicable ITU-R Recommendations.
	Successful Inter-Agency Communication Rate	≥99% of calls connect successfully to intended recipient.
Security	Encryption Usage Rate	100% of voice and data transmissions encrypted by default.
	Authentication Success Rate	99.9% of devices authenticated successfully.
	Over-the-Air Rekeying (OTAR) Success Rate	≥99% of OTAR processes completed successfully.
Operational Efficiency	Call Setup Time (Latency)	<500 ms for voice calls.
	Device Configuration Time	<5 min for remote device configuration changes.
	Equipment Robustness	Mean time to failure greater than 1 year.
	Battery life	>8 h of active operational use.

**Table 4 sensors-25-05156-t004:** Comparative overview of approved tactical MCX 4G/5G solutions in Romania.

Feature	Ericsson Tactical Bubble	Athonet HPE Tactical Cube	Nokia Tactical Solutions
**Core Network**	Ultra Compact Core (5G-ready)	Mobile 4G/5G Core	5G/4G with Nokia Perimeter
**Communication Services**	MCPTT, MCData, MCVideo	PTT, Data, Video, Network Slicing	PTT, Data, Video, Cell Edge Processing
**Deployment Flexibility**	Compact, Mobile, Scalable	Containerized Architecture	Rapid Deployment Mobile Network
**Interoperability**	Full 3GPP MCX Compliance	Interoperable with LTE/5G and MCX	Supports MCX, LTE, 5G, Edge Computing
**Network Management**	Ericsson Network Manager (Centralized)	Local and Remote Management	Nokia Network Manager with Edge Tools
**Use Cases**	Tactical, Military, Emergency Response	Military, Public Safety, Remote Ops	Defense, Disaster Response, Isolated Areas

**Table 5 sensors-25-05156-t005:** Comparative analysis of interoperable MCX application suites.

Feature	Team on Mission—Streamwide	Leonardo—Mission Critical Services	Nokia—Team Comms
**Solution Type**	MCX Suite for Mission Teams	MCX for Critical Service Operations	MCX Suite with LTE/5G Support
**Communication Services**	MCPTT, MCVideo, MCData	MCPTT, MCVideo, MCData	MCPTT, MCVideo, MCData
**Interoperability**	LTE/5G, Legacy System Integration	LTE/5G, Legacy System Integration	LTE/5G, Legacy System Integration
**Security**	End-to-End Encryption	End-to-End Encryption	End-to-End Encryption with Edge Focus
**Network Management**	Streamwide Centralized Management	Remote and Local Control Capabilities	Nokia DAC Management Suite
**Operational Domains**	Public Safety, Emergency Response	Defense, Critical Services	Defense, Public Safety, Interventions
**Performance**	Scalable over LTE/5G	Performance Optimization over LTE/5G	High Performance with Edge Computing

**Table 6 sensors-25-05156-t006:** IMT-2030 proposed capabilities.

Capability	Description	Research Target/Examples
Peak Data Rate	Maximum achievable data rate under ideal conditions per device.	50, 100, 200 Gbit/s
User Experienced Data Rate	Achievable data rate available ubiquitously across the coverage area to a mobile device.	300 Mbit/s, 500 Mbit/s
Spectrum Efficiency	Average data throughput per unit of spectrum resource and per cell.	1.5×–3× IMT-2020
Area Traffic Capacity	Total traffic throughput served per geographic area.	30 Mbit/s/m^2^, 50 Mbit/s/m^2^
Connection Density	Total number of connected and/or accessible devices per unit area.	10^6^–10^8^ devices/km^2^
Mobility	Maximum speed at which defined QoS and seamless transfer between radio nodes (multi-layer/multi-RAT) can be achieved.	500–1000 km/h
Latency	Contribution by the radio network to the time from when the source sends a packet to when the destination receives it.	To be defined
Reliability	Capability of transmitting a predefined amount of data within a predetermined time duration with a given probability.	1–10^−5^ to 1–10^−7^
Coverage	Ability to provide access to communication services for users in a desired service area, defined as the cell edge distance of a single cell.	To be defined
Positioning	Ability to calculate the approximate position of connected devices.	Accuracy: 1–10 cm
Sensing-Related Capabilities	Ability to provide functionalities like range/velocity/angle estimation, object detection, localization, imaging, and mapping.	Measured in terms of accuracy, resolution, detection rate, false alarm rate, etc.
AI-Related Capabilities	Ability to support AI-enabled applications through functionalities like distributed data processing, learning, computing, model execution, and inference.	To be defined
Security and Resilience	- Security: Preservation of confidentiality, integrity, and availability of information and protection against cyberattacks. - Resilience: Ability to operate correctly during and after disturbances.	To be defined
Sustainability	Ability to minimize greenhouse gas emissions and environmental impacts throughout the lifecycle, focusing on energy efficiency, resource optimization, and equipment longevity.	Energy efficiency: bits transmitted/received per unit of energy (bit/Joule).
Interoperability	Radio interface based on member inclusivity and transparency to enable functionality between different system entities.	To be defined

**Table 7 sensors-25-05156-t007:** Bridging to 6G in 3GPP Rel-19.

Main Feature	Source Document(s)	Scope/Focus Area
NR MIMO for Phase 5	RP-234007 (NR MIMO Phase 5)	Continues Multiple In, Multiple Out (MIMO) mobility specifications.
Evolution of NR Duplex operation: Sub-Band Full Duplex (SBFD)	RP-234035 (Evolution of NR Duplex Operation: SBFD)	Specifies BS RF conformance, RRM performance, and BS/UE demodulation performance requirements to support SBFD operation.
Artificial Intelligence (AI)/Machine Learning (ML) for NR Air Interface	RP-234039 (AI/ML for NR Air Interface) RP-234054 (Study on AI/ML for NG-RAN) RP-234055 (Study on AI/ML for Mobility in NR)	Wireless AI for device mobility enhancements by three projects in Release 19: study on AI/ML for Next-Gen Radio Access Network (RAN 3 led), study on AI/ML to enhance 5G NR mobility (RAN 2 led), and Work on AI/ML Air Interface (RAN 1 led), also by beam management and precise positioning.
Low-Power Wake-Up Signal and Receiver for NR (LP-WUS/WUR)	RP-234056 (LP-WUS/WUR)	Introduces methods to increase energy efficiency driven by a new energy-saving design, dedicated to small IoT devices (such as sensors and wearables).
Enhancements of Network Energy Savings for NR	RP-234065 (Network Energy Savings Enhancements)	Introduces new techniques to improve network energy savings, extending power efficiency innovations to the network.
NR Mobility Enhancements for Phase 4	RP-234036 (NR Mobility Enhancements Phase 4)	Continues 5G device mobility enhancements, and enhances measurements for Layer 2 mobility, and conditional mobility with short interruption.
Non-Terrestrial Networks (NTN) for NR Phase 3	RP-234078 (NTN for NR Phase 3); RP-234077 (NTN for IOT Phase 3)	Contains enhancements to NR-NTN for ubiquitous broadband access (5G NR-NTN for complementary terrestrial networks in underserved areas) and IoT-NTN for global IoT connectivity (5G IoT-NTN for addressable market expansion for massive 5G IoT).
XR (eXtended Reality) for NR Phase 3	RP-234080 (XR (eXtended Reality) for NR Phase 3)	XR evolution by delivering 5G-enhanced XR experiences in terms of system efficiency and user experience, and by delivering unlimited and enhanced XR experiences.
Data collection for SON (Self-Organizing Networks)/MDT (Minimization of Drive Tests) in NR Standalone and MR-DC (Multi-Radio Dual Connectivity) for Phase 4	RP-234038 (SON/MDT for NR/MR-DC Phase 4)	Enhances mobility robustness optimization (MRO), and enhanced SON/MDT for new services focuses on new services including intra-non-terrestrial (NTN) network mobility and network slicing.

**Table 8 sensors-25-05156-t008:** NTN key characteristics.

Characteristic	Pre-5G	5G	6G (Projected)
TN-NTN Integration	Separate optimization	NTN integrated with minimal TN impact	Full joint optimization
Satellite Access	Limited	Supported	Native integration
Architecture	Separate TN and NTN	Primarily TN-based	Unified MD-ML-MB NTN
Flexibility	Limited	Improved	Highly flexible and reconfigurable
Standardization	Separate	Collaborative	Fully integrated approach
Operator Collaboration	Limited	Increased	Extensive MNO-SNO collaboration
Air Interface	Separate for TN and NTN	Adapted for NTN	New specifications for joint TN-NTN
Business Models	Traditional	Evolving	New models required

**Table 9 sensors-25-05156-t009:** Other standardization activities of AI/ML frameworks apart from 3GPP.

Standard Body	Document/Series	Scope/Focus Area
3GPP SA5 (Mgmt)	ETSI TS 128 105 V17.4.0 (2023-07) [189] ETSI TR 128 908 V18.0.0 (2024-05) [190] 3GPP TS 28.105 V17.10.0 (2025-01) [191] 3GPP TS 28.105 V18.7.0 (2025-03) [192] 3GPP TS 28.105 V19.2.0 (2025-03) [193]	Management and orchestration capabilities and services for 5G systems that use AI/ML.
3GPP RAN (Radio)	3GPP TR 37.817 V17.0.0 (2022-04) [194] ETSI TR 128 908 V18.0.0 (2024-05) [190]	Principles for AI-enabled RAN intelligence, AI functionality, AI-enabled optimization, use cases, and provides solutions for AI-enabled RAN.
3GPP SA1/SA6	3GPP TS 22.261 V15.9.0 (2021-09) [195] ETSI TS 122 261 V16.14.0 (2021-04) [196] ETSI TS 122 261 V17.11.0 (2022-10) [197] 3GPP TS 22.261 V16.18.0 (2025-06) [198] 3GPP TS 22.261 V17.15.0 (2025-06) [199] 3GPP TS 22.261 V18.18.0 (2025-06) [200] 3GPP TS 22.261 V19.11.0 (2025-06) [201] 3GPP TS 22.261 V20.3.0 (2025-06) [31]	AI/ML types of operations, KPI for AI/ML model transfer in 5G systems.
ITU-T SG13	Y.3172 (06/19), 11.1002/1000/13894 [202]	Architectural framework for integrating ML into future networks, high-level architecture on an IMT-2020, ML pipeline overlay network.
ITU-T SG13	Y.3173 (02/20), 11.1002/1000/14133 [203]	Evaluation of Networks Intelligence. Specifying a framework and method for evaluating future networks intelligence, including IMT-2020, and identifying representative use cases.
ITU-T SG13	Y.3174 (02/20), 11.1002/1000/14134 [204]	Data Handling. Description of a generic framework for data management that will enable ML in future networks, including IMT-2020, and examples of its implementation on specific underlying networks.
ITU-T SG13	Y.3176 (09/20), 11.1002/1000/14402 [205]	ML Marketplace Architecture. Providing high-level architecture requirements for integrating machine learning (ML) markets into future networks, including IMT-2020.
ITU-T SG13	Y.3181 (09/22), 11.1002/1000/15058 [206]	AI Sandbox. Providing an architectural framework for a high-level architecture for the ML sandbox used in future networks, including IMT-2020.
ITU-T SG13	Y.3061 (12/23), 11.1002/1000/15735 [207]	Architectural requirements, component descriptions, and associated sequence diagram specifications needed in the design of an architectural framework for autonomous networks.
ITU-T SG13	Y.3142 (04/24), 11.1002/1000/15869 [208]	Using AI/ML technologies to improve network design mechanisms, specifying ways in which AI/ML can be integrated to optimize network capacity design and topologies.
O-RAN WG2	O-RAN.WG2.AIML-v01.03 (2021-10) [209] ETSI TS 103 983 V3.1.0 (2024-01) [210]	AI/ML lifecycle management, including model design, data composition and access during model runtime, and model deployment solutions. AI/ML models use algorithms to process data by analyzing past and current data events, making it easier to find patterns that help eliminate human error.
O-RAN WG2	O-RAN.WG2.TS.Non-RT-RIC-ARCH-R004-v07.00 (2025-06) [211] ETSI TR 104 037 V12.0.0 (2025-04) [212]	The architecture of the real-time RAN intelligent controller (Non-RT RIC), presenting a Non-RT RIC architecture diagram, and providing requirements for the Non-RT RIC framework, Non-RT RIC logical functions and services of the R1 interface. The functionalities and services of the Non-RT RIC framework exposed to applications.
O-RAN WG2	O-RAN.WG2.TS.A1GAP-R004-v05.01 (2025-06) [213] ETSI TS 103 983 V4.0.0 (2025-05) [214]	Presentation of general aspects and principles applied in the A1 interface within the O-RAN architecture.

## Data Availability

Not applicable.

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
