# Peer review of "Mission-Critical Services in 4G/5G and Beyond: Standardization, Key Challenges, and Future Perspectives"

_sensors, 2025, doi:10.3390/s25165156_

Round 1

Reviewer 1 Report

Comments and Suggestions for Authors

In this manuscript, the authors provide a comprehensive review of Mission-critical services (MCX), including the recent advancements from the standardization organizations. Initially, the authors present the standardization efforts (recent, ongoing and upcoming) towards MCX from 3GPP, O-RAN, ETSI groups, as well as GCF and the BroadEU.net initiative. Then, they are discussing the different types of communication that are included in MCX, namely the MCPPT, MCdata and MCvideo, detailing the relevant technical enablers (eg., ProSe, slicing, etc.) and specific requirements. Moreover, they illustrate how MCX can be used in verticals, such as PPDR, Transportation, power grids and resource industries and list existing MCX deployments and implementations. Finally, the authors focus on ongoing research and provide future research directions, mainly related to the challenges that will be addressed in the 6G era. The survey paper is interesting and relevant to the topic of the journal. Some minor comments should be addressed before publication to enhance the quality of this work:

  • There are many typos and syntax errors, e.g., line 174, line 191, line 265, line 363, line 571, etc. Please carefully proofread your manuscript.
  • The quality of most Figures needs to be improved, especially for Figures 3 – 9.
  • Some technical enablers that are intended to play a significant role in 6G (e.g., network slicing) are only described superficially. Please elaborate on these enablers and also describe more extensively the type of KPIs that can be used to monitor the performance of the MCX and also the Service Level Agreements that can be used to guarantee MC services.
  • Special description must be provided for KPIs that are related to disaster in the PPDR use cases, for instance related to network restoration time or latency and reliability of the network during the emergency to cover first responders.
  • In the future trends section, the AI/ML subsection requires additional details. First, several other standardization activities of AI/ML frameworks apart from 3GPP that are designed need to be mentioned, e.g., O-RAN AIMLF and ITU-T ML framework for supporting the management functions. Moreover, a brief description of the type of AI/ML algorithms that are used in these scenarios should be included (right before the Conclusions sections). Are the AI/ML algorithms based on deep learning? Are they based on reinforcement learning? Please elaborate on these issues.

Author Response

Dear Reviewer,

Thank you very much for your valuable feedback and insightful suggestion. We appreciate the time and effort you have taken to review our work, and we believe your input will significantly improve the quality of the paper.

All your observations have been addresed. Please find in the attached file the responses to each comment individually.

Reviewer 2 Report

Comments and Suggestions for Authors

This paper presents the first panoramic review of the three MCX services in 4G/5G standards, technologies, industry use cases, and 6G outlook, offering the latest normative progress and a tactical-grade product comparison, and identify NTN global coverage and AI-native intelligence as the two key enablers for reliability, resilience, and automation in 6G. Overall, the survey is comprehensive, yet some issues remain:

1.The NTN section highlights LEO/HAPS coverage and low latency, but under-discusses practical challenges such as spectrum coexistence, rain fade, regulatory constraints, and terminal power consumption.  

2.Although AI-native communications mentions XAI and model security, it does not quantify the signaling overhead, model-training latency, or terminal compute requirements introduced by AI.

Author Response

(The authors gave the same response as above.)

Reviewer 3 Report

Comments and Suggestions for Authors
  1. The abstract's statement "5G networks provide the essential building blocks" is too general. Please clarify the capabilities these "building blocks" refer to with specific metrics.
  2. Figure 3 uses numerous abbreviations but lacks captions. It is recommended that the full abbreviations be provided to facilitate understanding for non-specialist readers.
  3. Section 2.4.1 discusses public safety scenarios at length and could be streamlined to two paragraphs, highlighting the content directly related to MCX performance requirements.
  4. Please standardize terminology throughout the document: MCData and MCVideo have inconsistent capitalization in some paragraphs and require revision according to 3GPP specifications.
  5. The conclusion focuses too much on the vision for 6G. It is recommended that a detailed summary of the frozen features in Release 19 be added.
  6. Please provide two additional references such as [A] and [B] on IoT modulation techniques to support the discussion of low-power communications in the resource sector in Section 2.4.4:

[A] DOI: 10.1109/TCOMM.2024.3511707

[B] DOI: 10.1109/TVT.2025.3567450

  1. Many passive voice sentences are too long. We recommend using active voice as appropriate to improve readability, for example, changing “is expected to be enhanced” to “will enhance.”

Author Response

(The authors gave the same response as above.)

Round 2

Reviewer 2 Report

Comments and Suggestions for Authors

No further comments.

Reviewer 3 Report

Comments and Suggestions for Authors

All of my comments have been addressed.